# `NeuralSolver`: Learning Algorithms For Consistent and Efficient Extrapolation Across General Tasks

**Bernardo Esteves**\*
INESC-ID,
Instituto Superior Técnico,
Universidade de Lisboa

**Miguel Vasco**
Department of Intelligent Systems
KTH Royal Institute of Technology
Stockholm, Sweden

**Francisco S. Melo**
INESC-ID,
Instituto Superior Técnico,
Universidade de Lisboa

## Abstract

We contribute `NeuralSolver`, a novel recurrent solver that can efficiently and consistently extrapolate, i.e., learn algorithms from smaller problems (in terms of observation size) and execute those algorithms in large problems. Contrary to previous recurrent solvers, `NeuralSolver` can be naturally applied in both same-size problems, where the input and output sizes are the same, and in different-size problems, where the size of the input and output differ. To allow for this versatility, we design `NeuralSolver` with three main components: a recurrent module, that iteratively processes input information at different scales, a processing module, responsible for aggregating the previously processed information, and a curriculum-based training scheme, that improves the extrapolation performance of the method. To evaluate our method we introduce a set of novel different-size tasks and we show that `NeuralSolver` consistently outperforms the prior state-of-the-art recurrent solvers in extrapolating to larger problems, considering smaller training problems and requiring less parameters than other approaches. Code available at `https://github.com/esteveste/NeuralSolver`

## 1 Introduction

Humans can solve complex reasoning tasks by *extrapolating*: employing and combining elementary logical components to build more elaborate strategies. Machine learning models excel at pattern recognition, often outperforming humans in classification [1, 2], control [3] and prediction tasks [4]. However, these models still struggle at reasoning , which affects their ability to maintain their performance for increasingly harder versions of the same task [5]. A particular type of difficulty emerges from the increase in *dimensionality* of the input. Intuitively, learning algorithms to perform tasks, e.g., finding the goal in a maze (Figure 1), becomes increasingly harder for larger problems.

We focus on designing learning algorithms that are able to efficiently extrapolate, i.e., are able to learn algorithms to solve tasks over small problems (in terms of the dimensionality of the input) and execute over arbitrarily large problems, without a significant loss in performance nor additional fine-tuning. Recently, a novel class of methods, which we denote by *recurrent solvers*, have been proposed that employ recurrent neural networks to extrapolate [5, 6]. Compared to feed-forward networks, which have finite depth, recurrent networks can be iterated to an arbitrary depth at execution time. By doing so, these models can extrapolate to significantly larger problems than the ones seen during training by executing more recurrent steps. However, current recurrent solvers can only be applied to *same-size* problems, such as the case of image generation, where the dimensionality of the input (e.g., image size) is the same as the dimensionality of the output. This limitation inhibits the use of these methods in a wide range of tasks, which we call *different-size* tasks, where the input and output dimensions are different (e.g., classification and decision-making tasks).

---

\*Correspondence to: `bernardo.esteves@tecnico.ulisboa.pt`

38th Conference on Neural Information Processing Systems (NeurIPS 2024).

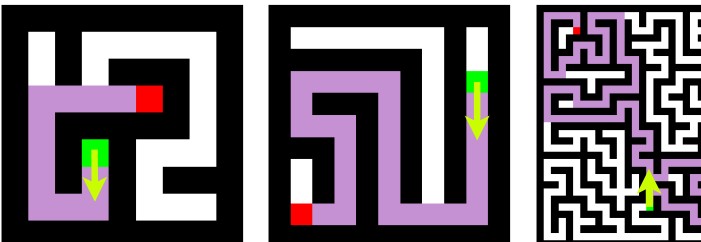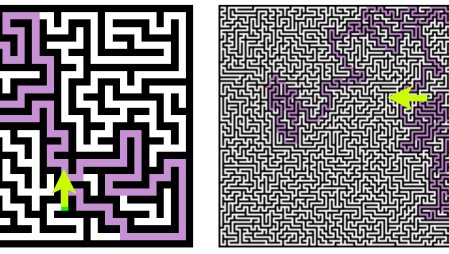

Figure 1: Observations of the 1S-Maze environment of sizes 7×7, 11×11, 33×33 and 129×129, where the agent (green square) must go to the goal (red square). The light green arrow represents the next target action that the model needs to predict, while the purple path represents the sequence of actions required to solve the maze.

We address this limitation and propose a novel recurrent solver that is able to consistently and efficiently extrapolate regardless of the size of the output and the type of problem. We name our approach `NeuralSolver`, a novel architecture for general tasks (same-size and different-size). Our approach introduces several architectural changes to recurrent solvers: a recurrent convolutional block that, through multiple iterations, perceives information at different scales in the input image, and a processing block (with an aggregation function) that merges information for the output of the model. Furthermore, we introduce a curriculum-based training scheme to improve the extrapolation abilities of our model. We train `NeuralSolver` on small observations (e.g., $15 \times 15$ images) using standard supervised learning techniques and apply the learned algorithm at test time with arbitrarily large observations (e.g., $256 \times 256$ images, or larger), with minimal loss in performance.

We evaluate `NeuralSolver` against prior recurrent solver in literature-standard same-size tasks and in a novel set of different-size tasks. We show across all tasks that our model significantly outperforms previous approaches in *extrapolation capability*, being able to execute learned algorithms in larger problems without loss in performance; in *training efficiency*, being able to learn algorithms from smaller problems; and in *parameter efficiency*, requiring $90\%$ less parameters than similar approaches.

In summary, the contributions of our work are:

- `NeuralSolver`: We propose a novel recurrent solver that uses a recurrent convolutional module, a processing module (with an aggregation function) and curriculum-learning to train algorithms in small problems and extrapolate them to arbitrarily large problems. Contrary to previous recurrent solver, our model can be used in same-size and different-size tasks;

- **Different-size tasks**: We contribute a novel set of different-size classification tasks for recurrent solvers, with images of arbitrary size and label information;

- **Consistent and efficient extrapolation**: We show that `NeuralSolver` significantly outperforms the prior recurrent solvers regarding extrapolation capabilities, training efficiency and parameter efficiency.

## 2   Related Work

Schwarzschild et al. [5] introduce a class of recurrent neural networks that can execute at test time more iterations than the number of iterations used during training, to solve problems more complex than the ones seen during training, which we denote by *recurrent-solvers*. To do so, they propose a recurrent network architecture based on residual neural networks (ResNets) [2] that shows logical extrapolation abilities across same-size tasks by leveraging the spatial invariances in the inputs. However, these networks suffer from *overthinking*, where the network deteriorates in performance when the number of iterations is extended beyond the training distribution. To solve the overthinking problem Bansal et al. [6] introduces two new components to the previous architecture: (1) a recall module, that integrates the input directly into specific layers of the recurrent network, safeguarding against potential loss or corruption of deep features; and (2) a progressive loss (PL) training scheme that incentivizes recurrent networks to iteratively refine the feature representation, preventing the network from memorizing the number of recurrent module applications. The authors show that their improved network is able to extrapolate to problems 16 times larger than the training size with

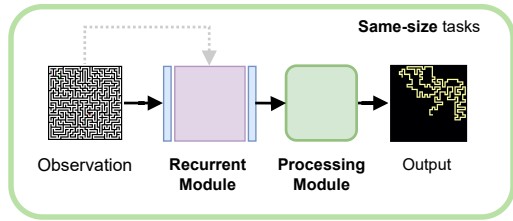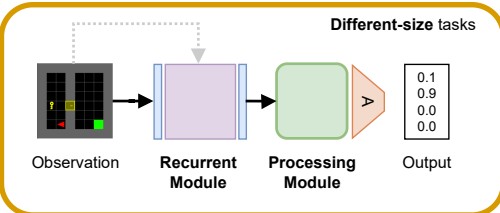

Figure 2: We design `NeuralSolver` with two fundamental components: (i) a *recurrent module* (purple), responsible for iteratively processing the input data regardless of its size; (ii) *processing module* (green), with an optional aggregation layer (A), responsible for generating the output and allowing our architecture to be used both in same-size and different-size tasks. Additionally, we employ a *curriculum-based* training scheme to improve the extrapolation performance of our architecture.

more than 95% accuracy on the same-size benchmark. We use this method as a baseline against our method. To address the problems of the computational efficiency and hyperparameter tuning of the progressive loss, [7] propose replacing the progressive loss with a delta loss term [8] and achieve similar performance to the original one. We propose a novel architecture that is able to outperform Bansal et al. [6] without the need of any additional loss to counter the overthinking problem (as shown in Appendix B.1). `NeuralSolver` is the first recurrent solver able to consistently and efficiently extrapolate learned algorithms in both same-size and different-size tasks.

## 3 The `NeuralSolver` Architecture

We address the challenge of designing a model that is able to efficiently learn algorithms from small problems that consistently extrapolate to larger problems. Contrary to previous recurrent solvers [5–7] we want to apply our model to both *same-size* tasks, where the input and output have the same dimensionality, and *different-size* tasks, where the input and output have different dimensionality. To address these challenges, we contribute `NeuralSolver`, depicted in Figure 2.

### 3.1 Model Architecture

We design our model with three components: (i) we extract information from the input data using a *recurrent module*, responsible for iteratively processing the input observation at different scales, allowing our model to cope with arbitrarily large input data; (ii) information is then sent to a *processing module*, responsible for aggregating (if necessary) the processed information and generating the output of the network, allowing our model to be used in both same-size and different-size tasks; (iii) to train our model we employ a *curriculum-based* scheme, that improves the extrapolation performance of `NeuralSolver` to larger problems.

**Recurrent Module**: This module consists of a layer normalized convolutional LSTM [9–11], a LSTM that uses convolutional layers instead of linear layers in its recurrent structure. The layer normalization is used to normalize the output of the convolutional layers and the cell state. We always provide the original input observations in the recurrent iterations of the LSTM, thus preventing the network from forgetting the input information over time, similar to the recall module in [6]. We maintain the dimensionality of the input and output of this module constant in order to perform multiple iterations over the data, allowing our model to process arbitrarily large observations. We show in Appendix B.1 that our recurrent module allows `NeuralSolver` to not suffer from overthinking, despite not employing any progressive loss during training. Moreover, in Appendix C.5 we compare the effect of different choices of recurrent architectures in the performance of our model.

**Processing Module**: This module consists of a block of three convolutional layers (following [6]) whose input is the last hidden state of the recurrent module. Their purpose is to reduce the number of channels to a desired output number of channels. Additionally, we have an optional aggregation layer (in our case a global max-pooling layer) that can be employed in different-size tasks to reduce the variable input size of the network to a fixed output size. For example, for a two-dimensional input observation, we first reduce the input to a $1 \times 1 \times o$ tensor, where $o$ is the desired output number of channels, and subsequently flatten the output tensor. This design allows our architecture to be used

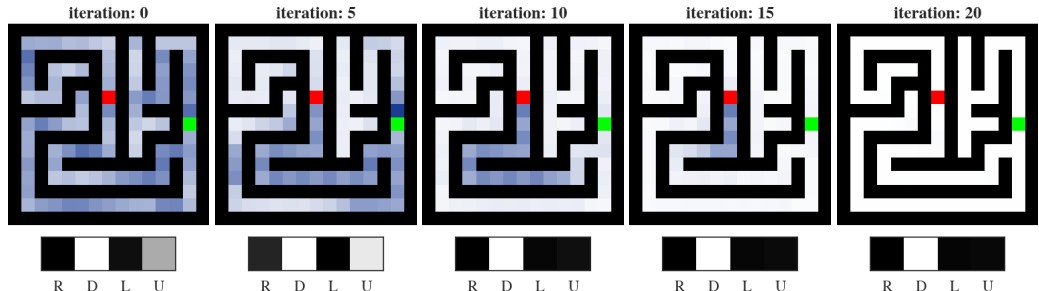

Figure 3: Propagation of information in `NeuralSolver` in a maze-like environment: the goal is for the agent (green) to find the goal position (red). Top: the difference between the value of the internal state of the recurrent module at each iteration step and the value at the final iteration. Larger differences are shown in dark blue and smaller differences in white. Bottom: additionally, we show the action probabilities predicted by the processing module of the model at different iterations, where the agent can move right (R), down (D), left (L), or up (U).

both in same-size and different-size tasks, with minimum changes. In Section 5.3 we compare the performance of different aggregation layers.

**Curriculum Learning Training**: We train our model on a set of smaller-size observations and test the performance of the model with larger dimensional observations (all belonging to the same task). Additionally, we employ a curriculum learning approach in different-size tasks to counter the effect of the reduced training signal (due to the smaller output dimensionality) [12, 13]: we initially train the models on lower-dimensionality observations, and then gradually increase the dimensionality of the observations every N epochs. To reduce the risk of catastrophic forgetting, we sample a minibatch of observations with a previously seen dimensionality (chosen uniformly at random) with a 20% chance, following previous work [12]. In Section 5.3 we show the importance of the curriculum-based training scheme in the extrapolation performance of `NeuralSolver`.

We evaluate the role of each of these components in our overall performance in Section 5.3. For additional details on the complete architecture of the method please refer to Appendix B.

### 3.2 Propagation of Information in `NeuralSolver`

To understand how information is processed in the recurrent module of our approach, we focus on the value of the internal state of the convolutional LSTM as a function of the number of iteration steps performed. In Figure 3, we show a maze-like environment at different iteration steps, where the task is the find the next position in the optimal path between the current position (in green) and the goal position (in red). We highlight the difference between the value of the internal state of the convolutional LSTM at each iteration step and the value of the internal state at the final iteration step (not shown in the figure). As the number of iterations increases, we observe that a larger number of positions in the maze become white as, for those positions, the value of the recurrent state does not change anymore. The speed of this convergence depends on the *receptive field of the convolution*: for example, a single convolution with a kernel size of 3 can capture the information from the adjacent pixels, propagating the information forward in one direction for a single pixel. As such, by using a convolutional recurrent neural networks we can propagate information across the input image by performing an appropriate number of iterations, regardless of the size of the problem.

This propagation of information also influences the output prediction. In Figure 3 (bottom), we highlight that the prediction of the output label is uncertain before iteration #10. However, in this iteration, the hidden state near the start goal (in green) has converged to the final value and the algorithm becomes certain of the correct label (in this case, "Down"). We provide more examples of learned algorithms in Appendix E.

## 4   Evaluation

We evaluate `NeuralSolver` to demonstrate how it ourperforms previous state-of-the-art recurrent solvers accordingly to three main criteria: (i) *extrapolation capability*, the ability to execute learned

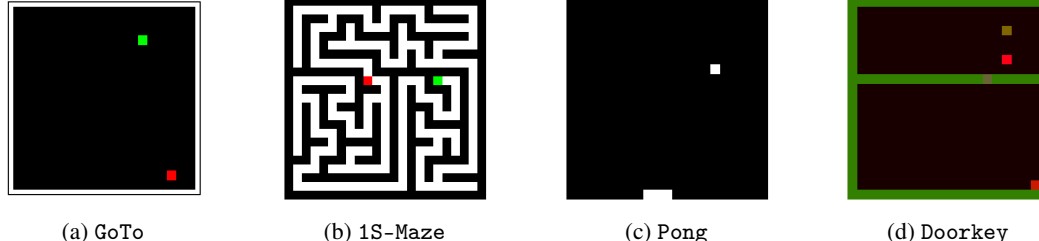

| (a) GoTo | (b) 1S-Maze | (c) Pong | (d) Doorkey |

Figure 4: We introduce a set of different-size classification tasks to evaluate the performance of recurrent solvers. In all tasks, the input is an image observation of the environment with arbitrary size. The output is an $n$-dimensional one-hot vector with: a) $n = 4$; b) $n = 4$; c) $n = 3$; d) $n = 4$.

algorithms in larger problems; (ii) *training efficiency*, the ability to learn algorithms from smaller problems and still maintain a high-level of extrapolation capability; and (iii) the higher *parameter efficiency* of the model. We compare our approach against previous recurrent solver baselines (Section 4.3), considering both same-size and different-size problems (Section 4.2). In Appendix B.3 we present the architecture of our model and training hyperparameters used in our evaluation.

### 4.1 Methods

**Model Training**: We apply `NeuralSolver` in supervised learning scenarios with datasets of input images/features and output images/features, in the case of same-size tasks, or output labels, in the case of different-size tasks. As such, we train our models using cross-entropy loss on datasets with small image sizes and evaluate the classification performance of our models in datasets with larger input images. In different-size tasks this approach is similar to typical imitation learning [14].

**Model Checkpoint Selection**: We consider a training and validation set split of 80% and 20%, respectively. To select the model checkpoint to be used for evaluation, we use the one that has the best performance on the validation set. In case of ties we select the later model checkpoint.

**Evaluation Metrics**: In our evaluation we run the model for a large number of iterations (detailed in Appendix B.3) and consider the best accuracy obtained by the models at any iteration [6]. We present the average best accuracy (in percentage) accompanied with the standard deviation. Additionally, we use the Almost Stochastic Order (ASO) test for statistical significance [15, 16], using a significance level of $\alpha = 0.05$ and employing the Bonferroni correction [17] for multiple comparisons. For more details on the ASO test and the statistical significance results, we refer the reader to Appendix D.

### 4.2 Scenarios

For same-size tasks we use the benchmark recently proposed by Schwarzschild et al. [5], consisting of three tasks: a logical toy task (`Prefix-Sum`), a maze-solving task (`Maze`), and a chess puzzle task (`Chess`). Additionally, we modify the Maze task to allow smaller size inputs, which we denote by `Thin-Maze`. For a detailed description of these tasks we refer the reader to the original paper.

Due to the lack of a literature-standard set of different-size tasks for algorithm extrapolation, we introduce four new classification scenarios. These new scenarios consider as input images of arbitrary sizes and as output vectors of fixed dimensions (more details in Appendix A):

`GoTo` (Figure 4a): Inspired by the Minigrid environment [18], the goal of the task is to select the action that moves the agent (in green) closer to the exit position (in red). The classification target is a four-dimensional one-hot vector, corresponding to the available actions (up, down, left, right).

`1S-Maze` (Figure 4b): Inspired by the Maze environment from the same-size task benchmark, the objective of the task is to select the action that moves the agent (in green) closer to the goal position (in red). The classification target is a four-dimensional one-hot vector, corresponding to the available actions (up, down, left, right). The name is an abbreviation of "1 Step Maze".

`Pong` (Figure 4c): Inspired by the Atari Pong game, the goal of the task is to select the action that moves the paddle horizontally to be underneath the ball. The classification target is a three-dimensional one-hot vector, corresponding to the available actions (left, right, no action).

Table 1: Extrapolation accuracy on the same-size tasks benchmark proposed in Schwarzschild et al. [5] and the `Thin-Maze` environment, with corresponding training and evaluation sizes $(s_t, s_T)$. Higher is better. All results are averaged over 10 randomly-selected seeds. We highlight the best average results. We use (†) to indicate stochastic dominance ($\epsilon_{\min} = 0$) and (*) to indicate almost stochastic dominance ($\epsilon_{\min} < 0.5$) of `NeuralSolver` over the baseline. We evaluate Bansal et al. [6] trained with progressive loss (PL) and without it.

| Model | Prefix-Sum $(32, 512)$ | Maze $(24, 124)$ | Thin-Maze $(11, 61)$ | Chess $(8, 8)$ |
|---|---|---|---|---|
| NeuralSolver | **100.00** $\pm$0.00 | **100.00** $\pm$ 0.00 | **99.97** $\pm$ 0.09 | **84.30** $\pm$0.40 |
| Bansal et al. [6] | 99.78 $\pm$0.78 * | 86.63 $\pm$28.21 * | 46.78 $\pm$37.49 † | 79.99 $\pm$3.27 † |
| Bansal et al. [6] + PL | **100.00** $\pm$0.01 * | 91.13 $\pm$27.37 * | 42.76 $\pm$37.56 † | 82.96 $\pm$0.19 † |
| FeedForward | 0.00 $\pm$0.00 † | 0.00 $\pm$ 0.00 † | 0.00 $\pm$ 0.00 † | 76.95 $\pm$0.35 † |

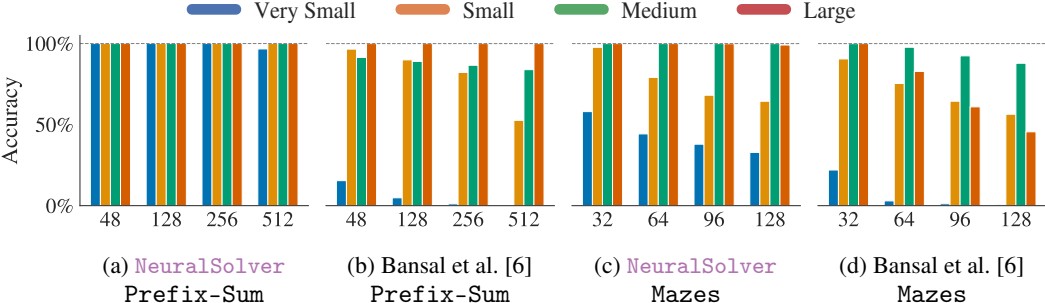

(a) NeuralSolver Prefix-Sum   (b) Bansal et al. [6] Prefix-Sum   (c) NeuralSolver Mazes   (d) Bansal et al. [6] Mazes

Figure 5: Training efficiency of `NeuralSolver` and Bansal et al. [6] on same-size tasks: we present the accuracy of the learned algorithms on extrapolating to problems with different dimensionality (columns). Each color represents a different training size, specific to each task, detailed in Appendix A.3 In the dashed line we show the upper-bound on the performance.

`Doorkey` (Figure 4d): Inspired by the Doorkey environment in Minigrid [18], the goal of this multi-step task is to reach the goal position (red pixel on the bottom right) by first picking-up a key and using it to open a lock door. The classification target is a four-dimensional one-hot vector, corresponding to the available actions (forward, rotate right, grab, toggle).

## 4.3 Baselines

**Bansal et al.** [6]: The current state-of-the-art recurrent solver architecture, that combines a recurrent Resnet network with a recall module and progressive loss. We employ the original source code from the authors and suggested training hyperparameters (when available).

**FeedForward**: A non-recurrent feed-forward network, based on the ResNet architecture, that has a fixed number of layers. This baseline is used to show the importance of the recurrent module for extrapolation.

**Random**: A lower-bound baseline, that randomly samples a classification label. This model is used to help understand the worst-case performance of the models in different-size tasks.

As the baseline models cannot be applied directly in different-size tasks, we modify them by introducing a global pooling layer similar to that of our model.

## 5 Results

### 5.1 Same-size Tasks

**Extrapolation Capability**: In Table 1 we present the extrapolation accuracy of `NeuralSolver` against the baselines in same-size tasks [5]. `NeuralSolver` achieves state-of-the-art extrapolation

Table 2: Total parameter count of the used models on the same-size tasks benchmark proposed in Schwarzschild et al. [5] and the `Thin-Maze` environment, in the scale of millions of parameters.

| Model | Prefix-Sum | Maze | Thin-Maze | Chess |
|---|---|---|---|---|
| NeuralSolver | 0.168 | 0.053 | 0.053 | 0.699 |
| Bansal et al. [6] | 3.124 | 0.784 | 0.784 | 12.057 |
| FeedForward | 58.804 | 17.888 | 17.888 | 285.735 |

Table 3: Extrapolation accuracy on the different-size tasks with training and evaluation sizes $(s_t, s_T)$. We employ curriculum learning to train all methods, with training sizes indicated in the table. Higher is better. All results are averaged over 10 randomly-selected seeds. We highlight the best average results. We use (†) to indicate stochastic dominance ($\epsilon_{min} = 0$) of NeuralSolver over the baseline.

| Model | 1S-Maze $([7, 15]], 129)$ | GoTo $([6, 20], 128)$ | Pong $([6, 20], 128)$ | DoorKey $([6, 20], 128)$ |
|---|---|---|---|---|
| NeuralSolver | **100.00** $\pm 0.00$ | **100.00** $\pm 0.00$ | **100.00** $\pm 0.00$ | **100.00** $\pm 0.00$ |
| Bansal et al. [6] | 74.14 $\pm 2.60$ † | 64.32 $\pm 8.86$ † | 71.97 $\pm 10.70$ † | 97.13 $\pm 1.46$ † |
| FeedForward | 72.47 $\pm 0.85$ † | 56.82 $\pm 3.74$ † | 48.99 $\pm 11.25$ † | 79.22 $\pm 11.86$ † |
| Random | 29.74 $\pm 0.41$ † | 29.46 $\pm 0.40$ † | 37.87 $\pm 0.47$ † | 29.52 $\pm 0.58$ † |

performance on same-size tasks. In particular, our approach is able to achieve a 100% accuracy on the `Prefix-Sum` and `Maze` tasks, when evaluated on a scenario with observations 16 and 5 times bigger, respectively, than during training. On the more complex `Chess` task, our model still outperforms the other baselines, achieving an average accuracy of 84.30%. In Appendix B.1 we show that NeuralSolver also does not suffer from *overthinking*.

**Training Efficiency**: In Figure 5 we show the accuracy performance of the learned algorithm with our model when trained with different problem sizes. The results show that NeuralSolver outperforms Bansal et al. [6] in extrapolating when trained with smaller input observations, achieving consistent upper-bound performance when trained in problems with only 8-dimensional (`Prefix-Sum`) and 16-dimensional (`Mazes`) input observatons. The results for the other environments are in Appendix C.1.

**Parameter Efficiency**: In Table 2 we present the total parameter count of NeuralSolver against the baselines in same-size tasks [5]. The results show that our model requires less than 10% of the total parameters of the baselines.

## 5.2 Different-size Tasks

**Extrapolation capabilities**: In Table 3 we present the extrapolation performance of NeuralSolver against the baselines in different-size tasks. Our model also outperforms prior state-of-the-art recurrent solvers across all different-size tasks: our approach is able to achieve upper-bound performance when extrapolating to observation sizes 9 (`1S-Maze`) and 6 (`GoTo`, `Pong`, `Doorkey`) times larger than the ones provided during training. In Appendix C.7, we explore the setting of *extreme* extrapolation, evaluating our model on very large image sizes ($256 \times 256$ and $512 \times 512$). The results show that, even in such challenging conditions, NeuralSolver is able to consistently extrapolate without losing performance, while the baselines fail to do so.

**Training Efficiency**: In Figure 6 we show the performance of our model against the Bansal et al. [6] model with different training sizes. The results show that, while the previous state-of-the-art consistently under performs across most tasks on bigger test sizes, NeuralSolver is still able to learn algorithms with suitable extrapolation abilities (often close to the upper-bound performance) when trained with smaller observations. The observed gap of our method to upper-bound performance also highlights that the novel set of different-size tasks can be employed to benchmark the extrapolation performance of future recurrent solvers when training using only extremely small problems.

**Parameter Efficiency**: NeuralSolver employs the same architecture across all different-size tasks, with 0.23 million total parameters. This corresponds to a 92% reduction in the number of parameters

Table 4: Extrapolation accuracy of different ablated versions of `NeuralSolver` in the proposed different-size tasks. Higher is better. All results are averaged over 10 randomly-selected seeds. We highlight the best average results. We use (†) to indicate stochastic dominance ($\epsilon_{min} = 0$) and (*) to indicate almost stochastic dominance ($\epsilon_{min} < 0.5$) of our default model over the ablated versions.

| Model | 1S-Maze | GoTo | Pong | DoorKey |
|---|---|---|---|---|
| `NeuralSolver` | **100.00** $\pm$ 0.00 | **100.00** $\pm$ 0.00 | **100.00** $\pm$ 0.00 | **100.00** $\pm$ 0.00 |
| Use AvgPool | 81.96 $\pm$28.50 † | 97.00 $\pm$ 3.87 † | 92.86 $\pm$14.66 † | 51.86 $\pm$38.98 † |
| Use 5L | 78.00 $\pm$ 5.73 † | 84.16 $\pm$16.39 † | **100.00** $\pm$ 0.00 | 92.99 $\pm$ 7.62 † |
| No CL | 95.26 $\pm$ 7.29 † | 64.12 $\pm$23.50 † | **100.00** $\pm$ 0.00 | 97.23 $\pm$ 4.73 * |
| No LSTM | 81.87 $\pm$10.09 † | 67.28 $\pm$14.51 † | 76.12 $\pm$17.63 † | 93.68 $\pm$ 7.88 † |

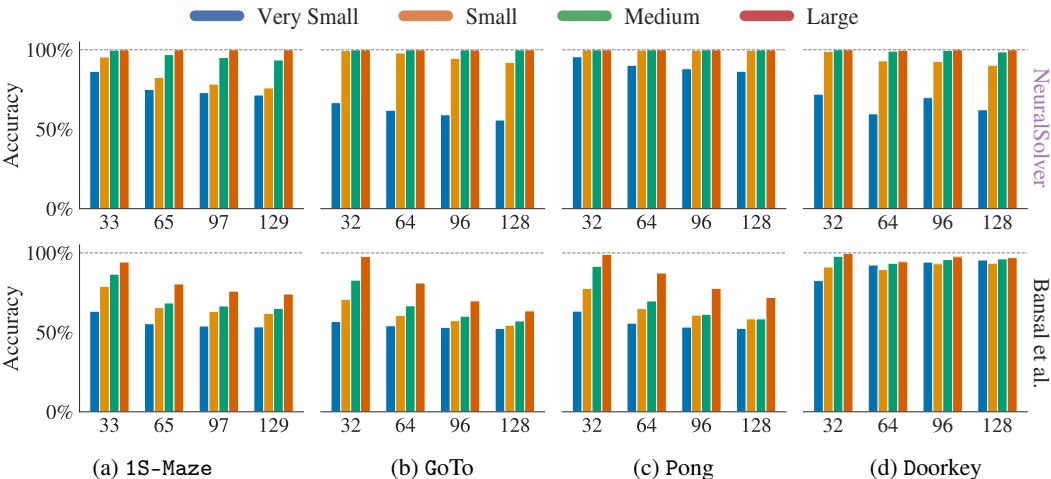

Figure 6: Training efficiency of `NeuralSolver` and Bansal et al. [6] on different-size tasks: we present the accuracy of the learned algorithms on extrapolating to problems with different dimensionality (columns). Each color represents a different training size, specific to each task, detailed in Appendix A.3. In the dashed line we show the upper-bound on the performance.

against Bansal et al. [6] (3.15 million parameters) and a reduction of 99.6% against the FeedForward baseline (71.57 million parameters).

## 5.3 Ablation Study

In Table 4 we perform an ablation study on the components of `NeuralSolver`, particularly on the role of: (i) the aggregation function, (ii) the depth of recurrent convolutional block, (iii) the training method and (iv) the type of recurrent layer in the convolutional block . For (i), we replace the global max-pooling layer of the processing module with an average-pooling layer (Use AvgPool). For (ii), we use 5 convolutional layers in the recurrent module (similar to Bansal et al. [6]) instead of the single convolutional layer (Use 5L). For (iii), we remove the curriculum-based training scheme (No CL). For (iv), we replace the LSTM with a ResNet block (No LSTM).

The results show that every component of `NeuralSolver` contributes to the overall extrapolation performance of the method. The use of an LSTM layer instead of a ResNet block in the recurrent convolutional module results in a significant improvement in performance. This result is aligned with previous works that have shown that gated-based recurrent neural networks empirically learn and generalize better than recurrent ResNets [19]. The use of max pooling as the aggregation function results in better performance than average pooling. Removing curriculum learning also results in a decrease in extrapolation performance of the method in some tasks (e.g., `GoTo`), highlighting the need to consider multiple (small) image sizes to extract relevant information during the training procedure.

Table 5: Average reward returns on the Minigrid Doorkey environment, with different sizes during execution. We show the average reward multiplied by $10^2$. We employ curriculum learning with sizes ($[6, 20]$) to train all models. Higher is better. We highlight the best average results. We use (†) to indicate stochastic dominance ($\epsilon_{min} = 0$) and (*) to indicate almost stochastic dominance ($\epsilon_{min} < 0.5$) of `NeuralSolver` over the baseline. All results are averaged over 10 randomly-selected seeds.

| Model | 20×20 | 32×32 | 64×64 | 128×128 |
|---|---|---|---|---|
| Oracle | 98.92 ±0.02 | 99.35 ± 0.01 | 99.69 ± 0.00 | 99.85 ± 0.00 |
| `NeuralSolver` | **98.82** ±0.26 | **99.12** ± 0.57 | **98.69** ± 2.44 | **98.02** ± 2.54 |
| Bansal et al. [6] | 98.47 ±0.48 * | 91.41 ±10.64 † | 43.09 ±31.18 † | 24.71 ±31.76 † |
| FeedForward | 96.14 ±1.63 † | 63.51 ± 7.63 † | 23.53 ± 7.34 † | 7.19 ± 3.18 † |

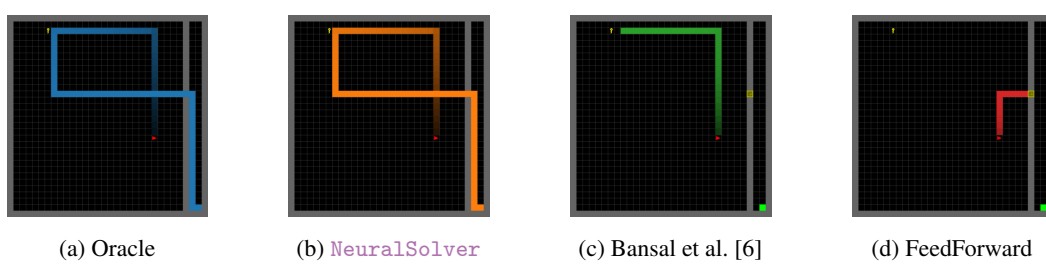

(a) Oracle        (b) `NeuralSolver`        (c) Bansal et al. [6]        (d) FeedForward

Figure 7: Example trajectories of the different methods when extrapolating to a Minigrid `Doorkey` environment with an image observation of size 64×64. The trajectory of the agents follows the gradient of the line (from darker to brighter). Additional examples in Appendix F.

We also observe a performance gain in using a single convolutional layer over a module composed of five layers (as used in [6]), despite requiring five times more the number of iterations to compensate.

### 5.4 `NeuralSolver` Allows Extrapolation on Sequential Decision-Making Tasks

We highlight the versatility of `NeuralSolver` by exploring visual imitation learning problems, in which, at each time-step, the algorithm is provided with an image of arbitrary size, and needs to output an action. These tasks can be solved through modern reinforcement learning methods [14, 20–26]. However, these methods lack the ability to *extrapolate*, i.e., are unable to learn algorithms (policies) on small problems, with lower-dimensional observations, and execute on larger problems, with higher-dimensional observations. We consider the `DoorKey` environment and train an imitation-learning policy algorithm using `NeuralSolver` with a dataset collected by a pretrained oracle agent. Subsequently, we employ the learned algorithm in the Minigrid `DoorKey` environment [18], directly as the agent's policy: at each time-step, mapping the observations of the agent (image of arbitrary size) into actions. We evaluate the performance of our method in extrapolating at execution time to larger observations against oracle policies specifically trained on each observation size.

The results in Table 5 highlight how `NeuralSolver` can achieve oracle-level performances in scenarios with larger observation sizes despite *never being trained on such conditions*. Moreover, the prior baselines struggle to maintain their performance for larger observations due to the accumulation of errors during task-execution, resulting in sub-optimal trajectories as shown in Figure 7.

## 6 Conclusion

We proposed `NeuralSolver`, a simple architecture that learns algorithms that can perform extrapolation across general tasks. We showed that our architecture consistently outperforms prior state-of-the-art recurrent solvers in regard to extrapolation performance, training efficiency and parameter efficiency. In future work, we plan on further improving the training efficiency of our method and explore it as an architecture for reinforcement learning to learn to perform sequential decision-making tasks in an online manner, while maintaining the ability to extrapolate.

## Acknowledgements

This work was partially supported by national funds through Fundação para a Ciência e a Tecnologia (FCT) with ref. UIDB/50021/2020 and the project RELEvaNT, ref. PTDC/CCI-COM/5060/2021. The first author acknowledges the FCT PhD grant 2023.02298.BD. This work has also been supported by the Swedish Research Council, Knut and Alice Wallenberg Foundation and the European Research Council (ERC-BIRD).

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

Table 6: Additional details on the datasets used in the evaluation of `NeuralSolver`. For the different-size tasks we show the dimensionality of the training examples used for curriculum learning.

| Task | Train size | Test Size | Training examples | Test examples |
|------|-----------|-----------|-------------------|---------------|
| GoTo | [6,8,10,12,15,17,20] | 128 | 50,000 | 10,000 |
| 1S-Maze | [5,7,9,11,13] | 121 | 50,000 | 10,000 |
| Pong | [6,8,10,12,15,17,20] | 128 | 50,000 | 10,000 |
| Doorkey | [6,8,10,12,15,17,20] | 128 | 50,000 | 10,000 |
| Prefix-Sum | 32 | 512 | 50,000 | 10,000 |
| Maze | 24 | 124 | 50,000 | 10,000 |
| Thin-Maze | 11 | 61 | 50,000 | 10,000 |
| Chess | 8 | 8 | 600,000 | 100,000 |

## A    Additional Details on the Evaluation Scenarios

We evaluate `NeuralSolver` on the same-size tasks benchmark introduced in Schwarzschild et al. [5] and on a novel set of different-size tasks. In Table 6 we show the training and test sizes for each task, as well as the number of training and test examples. For the different-size tasks, we employ curriculum learning during training and show the dimensionality of the training examples in the table.

### A.1    Different-size Tasks

Due to the lack of a benchmark for different-size tasks, we introduce a set of four classification tasks with arbitrary input (image) observation dimensionality and fixed output size (classification labels). For each task we generate a dataset, consisting of 50,000 training examples, with 80% used for training and 20% for validation. Additionally, we include 10,000 test examples for evaluation purposes, containing observations with a larger dimensionality than the training examples.

`GoTo`: Inspired by the Minigrid Environment [18], the `GoTo` task is a simple grid setting, with a 1-pixel white border, and 1-pixel green player, and a 1-pixel red goal, as shown in Figure 4a. The classification target is a one-hot vector of 4 dimensions, corresponding to the next-step actions available to the agent: up, down, left, right. To generate the dataset we use a simple path-finding algorithm from the player to the goal, where the agent starts by minimizing the vertical distance to the goal until it reaches the same vertical position, and then minimizes the horizontal distance until it reaches the goal.

`1S-Maze`: This task is based on the `Maze` task from [5]. The input is a maze with the player and goal positions, and the objective is to move the player (in green) to the goal (in red) position by solving the maze. We use the official dataset from the original paper, where we made the green player have a random start position so that it does not become trivial to solve. We also changed the thickness of the walls and the border to 1 pixel, as shown in Figure 4b. The classification target is a one-hot vector of 4 dimensions, corresponding to the next-step actions available to the agent: up, down, left, right. The dataset is generated using a depth-first search algorithm, where the label is the next action to take to reach the goal.

`Pong`: Inspired by the Atari game, the `Pong` task is a simplified version of the game, where the objective is to move the paddle horizontally to the ball position, as seen in Figure 4c. The classification target is a one-hot vector of 3 dimensions, corresponding to the next-step actions available to the agent: left, right, no action. The dataset is generated by a simple agent that follows the ball, keeping the paddle centered on the ball.

`Doorkey`: This task is inspired by the Minigrid Doorkey environment [18]. The goal is to move the player (red pixel on top) to the goal (red pixel on bottom right) through a sequence of steps, as seen in Figure 4d: moving to the key (yellow pixel), grabbing it, moving to the door (different color pixel on middle separator), opening it and finally moving to the goal. The classification target is a one-hot vector of 4 dimensions, corresponding to the next-step actions available to the agent: forward, rotate right, grab, toggle. We generate the targets using a simple oracle agent that solves the task. In the

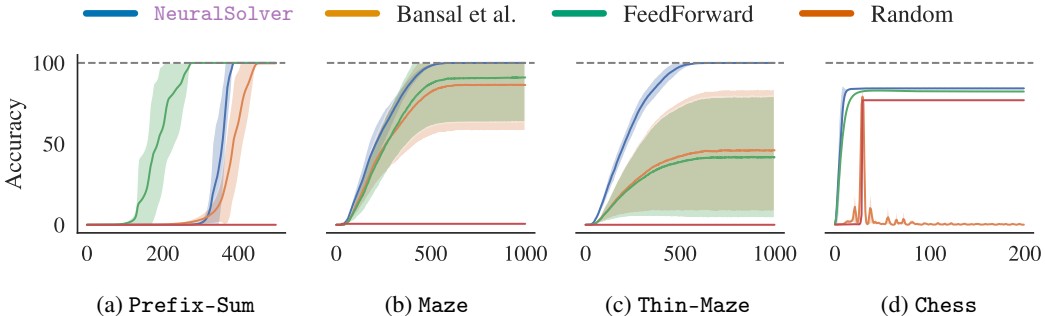

Figure 8: Extrapolation accuracy of the different methods as a function of the number of iterations in the same-size task benchmark. All results are averaged over 10 randomly-selected seeds.

original reinforcement learning environment, upon reaching the goal the agent receives a reward of $r = 1 - 0.9 * (\frac{t}{T})$, where $t$ is the number of steps of the agent and $T$ is the maximum episode length (if $t > T$ the agent receives no reward and the episode restarts). In conjunction with the oracle agent, we use this environment to evaluate the performance of the learned algorithm in the original environment.

## A.2 Same-size Tasks

We employ the same-size task benchmark introduced in [5]. We present a short description of the tasks used in the benchmark. For a more detailed description of the tasks, please refer to the original paper.

Prefix-Sum: This task consists of computing the prefix-sum module two of an input binary array, where the $i_n$ bit of the output is the module two sum of the current and the previous $n$ bits of the input.

Maze: This is a maze-solving task where the output is the path from the player to the goal. The training is done on mazes of size 24×24 pixels, and tested on mazes of size 124×124. While on the original paper, it is referred that the mazes have a training size of 9 and a test size of 59, each maze has a border of 3 pixels and the paths have a width of 2 pixels, thus $9 \times 2 + 3 \times 2 = 24$ and $59 \times 2 + 3 \times 2 = 124$.

Thin-Maze: Similar to the 1S-Maze different-size task, this task adapts the previous scenario and modifies the thickness of the walls and the border to 1 pixel. Thus the training and test sizes are 11×11 and 61×61, respectively.

Chess: The chess task comprises of chess puzzles, where the input is an 8×8×12 array indicating the position of the pieces on the board, and the output is an 8×8 binary array indicating the optimal move origin and position to solve the puzzle. Using the original dataset, the training involves 600,000 puzzles below a rating of 1,385, while testing employs 100,000 examples above this threshold.

## A.3 Training Efficiency Task Sizes

To simplify the interpretation, we omitted the training sizes of each task used in Figure 5 and Figure 6. In Table 7 we detail the training sizes of each task.

Table 7: Training sizes of each task used for the training efficiency evaluation

| Task | Very Small | Small | Medium | Large |
|---|---|---|---|---|
| Prefix-Sum | 4 | 8 | 16 | 32 |
| Maze | 16 | 20 | 24 | 28 |
| 1S-Maze | 9 | 11 | 13 | 15 |
| GoTo, Pong, Doorkey | 8 | 12 | 15 | 20 |

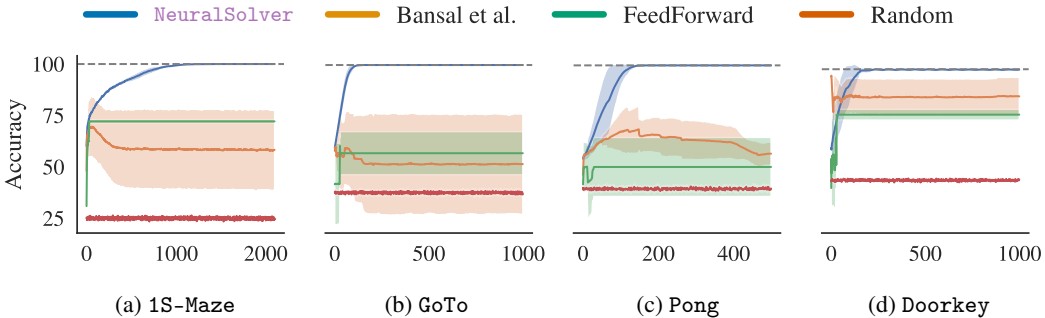

Figure 9: Extrapolation accuracy of the different methods as a function of the number of iterations in the different-size tasks. All results are averaged over 10 randomly-selected seeds.

# B  Additional Details on `NeuralSolver`

## B.1  `NeuralSolver` Does Not Suffer From Overthinking

We evaluate the accuracy performance of our model and the baseline models across all same-size (Figure 8) and different-size (Figure 9) tasks as a function of the number of iterations performed. The results show that `NeuralSolver` does not suffer from overthinking: the performance of the model remains constant as we increase the number of iterations.

## B.2  Changes We Tried That Did Not Improve `NeuralSolver`

We present a list of architectural changes we explored for `NeuralSolver` that did not bring any substantial improvement to the extrapolation performance of our method:

- Using batch normalization in the ResNet blocks of the recurrent module;
- Increasing or decreasing the warming up period of the training;
- Reducing the kernel sizes of the final output head;
- Having in the ResNet block one convolutional layer with $1 \times 1$ filters and the remaining ones with $3 \times 3$ filters;
- Reducing the amount of ResNet blocks in the original Bansal et al. [6] architecture from two to one;
- Removing the ResNet block, and using just a single recall convolution;
- Using an Tanh regularization in the output of the recurrent module, to avoid the drift or explosion of the recurrent memory;
- Using the progressive loss to improve overthinking problem in different-size tasks (did not help in standard Bansal et al. [6] and initial experiments with LSTMs)

## B.3  Model Implementation and Training Hyperparameters

In Table 8 we present the model implementation used across all tasks. In Table 9 we present the hyperparameters used for each task. In the same-size tasks we employ the same hyperparameters as in Bansal et al. [6], with the exception of the `Maze` task, where we increase the training to 150 epochs and keeping the decay schedule at epoch number 100, as present in the source code. We also removed the warm-up for all our models, since they brought no performance improvement to the models.

## B.4  Computational Complexity

We evaluate the computational complexity of the models, by computing the amount of Multiply-Add Operations (MACs) for a single training example across all tasks in Table 10, in the scale of billions or gigaMACs. The results show that `NeuralSolver` is more computationally efficient than Bansal et al. [6], requiring less than 30% of the operations per training example.

Table 8: Model implementation details of `NeuralSolver` across all tasks. For different-size tasks, the output size $o$ is variable. We employ the same architecture in the `Maze` and `Thin-Maze` environments.

| Dataset | Width | # Channels in Hidden Layers |
|---------|-------|------------------------------|
| Different-size Tasks | 64 | 64, 64, $o$ |
| Prefix-Sum | 100 | 400, 200, 2 |
| Maze | 32 | 32, 8, 2 |
| Chess | 128 | 32, 8, 2 |

Table 9: Training hyperparameter values for `NeuralSolver` across all tasks.

| Hyperparamter | 1S-Maze | Pong | GoTo | Doorkey | Prefix-Sum | Maze | Chess |
|---------------|---------|------|------|---------|------------|------|-------|
| Optim. | Adam | Adam | Adam | Adam | Adam | Adam | SGD |
| Learning Rate | 1e-3 | 2.5e-4 | 1e-3 | 1e-3 | 1e-3 | 1e-3 | 1e-2 |
| Decay Schedule | [100] | - | - | - | [60, 100] | [100] | [100, 110] |
| Decay Factor | 0.1 | - | - | - | 0.01 | 0.1 | 0.01 |
| Warm-Up | 0 | 0 | 0 | 0 | 10 | 10 | 3 |
| Epochs | 150 | 50 | 50 | 50 | 150 | 150 | 120 |
| Clip | 2.0 | 2.0 | 2.0 | 2.0 | 1.0 | - | - |
| Curriculum Epochs | 8 | 4 | 4 | 4 | - | - | - |
| Weight Decay | 2e-4 | 2e-4 | 2e-4 | 2e-4 | 2e-4 | 2e-4 | 2e-4 |
| Std Dropout (LSTM) | 0.3 | 0.3 | 0.3 | 0.3 | 0.3 | 0.3 | 0 |
| Gal Dropout [27] (LSTM) | 0.4 | 0.4 | 0.4 | 0.4 | 0.4 | 0.4 | 0 |
| Eval Iterations | 2000 | 1000 | 500 | 1000 | 500 | 1000 | 200 |
| Extreme Eval Iterations | 20000 | 2000 | 2000 | 2000 | - | - | - |

## B.5 Computational Resources

All experiments were executed on workstations with Nvidia RTX 3090 and RTX 4090 with 24GB of vram each. Training and evaluating (with a maximum size of 128) a single run takes less than 1 day to run in all tasks, with the exception of the chess task, where each run takes around 58 hours for the Bansal et al. [6] model and 26 hours for `NeuralSolver`.

## B.6 Comparison with Bansal et al. [6]

In this section we highlight the major differences between `NeuralSolver` and Bansal et al. [6], shown in Figure 10. The main architectural differences between our model and the prior Bansal et al. [6] architecture are: (i) the recurrent module used; (ii) the removal of the initial projection layer; and the design processing module(iii).

**Recurrent Module**: The recurrent module in `NeuralSolver` is a layernorm convolutional LSTM (Figure 11) where x is the input, h is the hidden state, c is the cell state of the LSTM. All convolutional layers in all models use the same length-three filters and padding scheme, without bias terms except for the convolutional layers in the LSTM block. Since the input is always the same across recurrent iterations, we pre-compute the convolution and layernorm passes, so that at each iteration we only sum the pre-computed value.

**Removal of Projection Layer**: We observed a small negative impact in extrapolation when using an initial projection layer for the recurrent module, and as such we removed it from the architecture. This results can be seen in Table 18 of Section C.6.

**Processing Module**: The processing module consists of three convolutional layers with widths specified in Table 8, with ReLU activations applied after the first two layers. The last convolutional layer produces two-channel outputs used for binary pixel classification in the case of the same input and output size tasks, or $N$ channel outputs in the case of fixed output size tasks, where $N$ is the number of possible outputs. Our contribution introduces an aggregation layer at the end of the model,

Table 10: Computational complexity (in gigaMACs) of different models across all tasks, measured in terms of the amount of Multiply-Add Operations necessary to run a single training example.

| Model | 1S-Maze | GoTo | Pong | Doorkey | Prefix-Sum | Maze | Thin-Maze | Chess |
|---|---|---|---|---|---|---|---|---|
| NeuralSolver | 4.12 | 9.76 | 9.76 | 9.76 | 0.62 | 3.39 | 0.71 | 5.74 |
| Bansal et al. [6] | 15.93 | 37.71 | 37.71 | 37.71 | 3.00 | 13.48 | 2.83 | 23.05 |
| FeedForward | 15.93 | 37.71 | 37.71 | 37.71 | 3.00 | 13.48 | 2.83 | 23.05 |

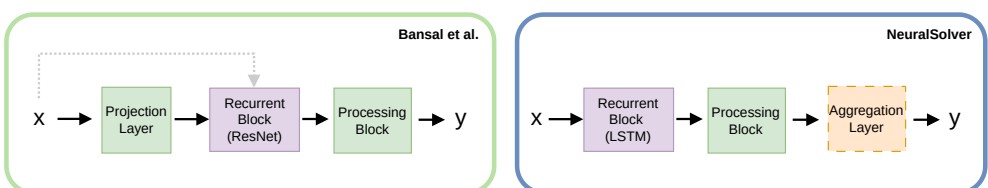

Figure 10: Simplified comparison of the architectural differences between NeuralSolver and Bansal et al. [6].

used for tasks with a different a target size from the training size. The aggregation layer consists of an adaptive max pooling that reduces the entire input size to 1, with exception for the input channels that are kept the same.

A complete description of the final architecture can be described as follows: for same-size tasks, given an input image $o$, we pass the input through a single layer ConvLSTM ($q$) to obtain the next hidden state ($h_1$ and cell state ($c_1$), $h_1, c_1 = q(o, h_0, c_0)$. The initial hidden ($h_0$) and cell ($c_0$) states are initialized at zeros. The recurrent process is given by feeding back the previous hidden and cell states to the LSTM, $h_t, c_t = q(o, h_{t-1}, c_{t-1})$. After a fixed size of recurrent iterations, we can obtain the model prediction at timestep $t$ by passing the hidden state $h_t$ through the processing module. The processing module is composed of three convolutional layers with 3x3 kernels denoted by $W_1, W_2, W_3$ and ReLU activation $\gamma$. As such, we perform $p_t = W_3 * \gamma(W_2 * \gamma(W_1 * h_t))$, and a final softmax activation ($\sigma$), $\hat{y}_t = \sigma(p_t)$. For different size tasks, before the softmax we perform a global maxpool operation ($G$) that reduces the processing output height and width to 1, $\hat{y}_t = \sigma(G(p_t))$.

## C    Additional Results

### C.1    Extended Training Efficiency on Same-size Tasks

In Figure 12 we show the performance of the learned algorithm with our model when trained with different training sizes on the Thin Maze and Chess tasks. The results show that only NeuralSolver outperforms Bansal et al. [6] in extrapolating when trained with smaller input observations on Thin Mazes. On the other hand, in the chess task, both models perform very similarly, with Bansal et al. [6] being slightly more performant on easier puzzle data, and NeuralSolver being slightly better with more data.

### C.2    Training and Validation Accuracy on Different-size Tasks

In Table 11 and Table 12 we show the training and validation accuracy for all models in the different-size tasks. The results show that all models are trained to optimal performance.

### C.3    Fine-tuning of the Progressive Loss Parameter

The Bansal et al. [6] model introduces a progressive loss term to help the model extrapolate. However, this term has an alpha ($\alpha$) parameter that needs to be tuned to each individual task. In Table 13 we show the results of the fine tuning of the alpha parameter for the different-size tasks. For each different-size task we select and present the best-performing alpha value (in bold). For the Thin-Maze task, we selected both the $\alpha = 0$ and the $\alpha = 0.1$ for the same-size benchmark comparison. For the remaining same-size tasks, we employ the author's suggested values for $\alpha$.

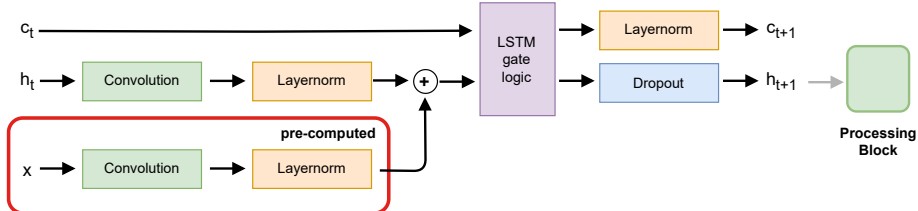

Figure 11: Diagram of layernorm convolutional LSTM used in `NeuralSolver`. After a desired amount of recurrent iterations, we use the hidden state `h` as input to the processing module, to obtain the output.

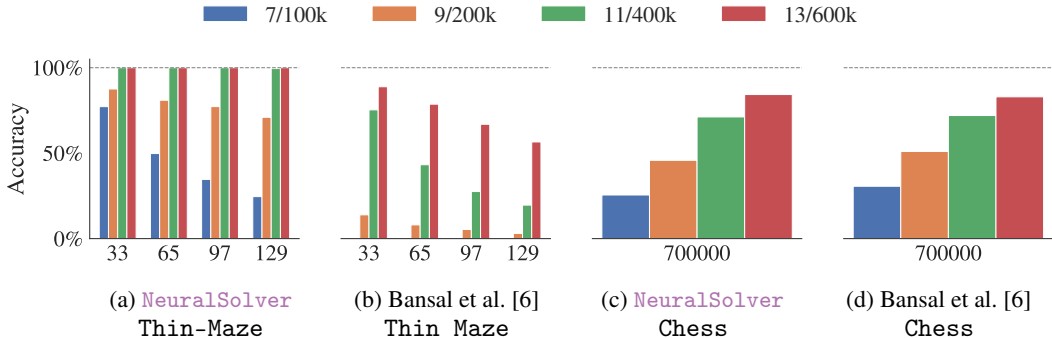

(a) `NeuralSolver`     (b) Bansal et al. [6]     (c) `NeuralSolver`     (d) Bansal et al. [6]
     `Thin-Maze`        `Thin Maze`          `Chess`           `Chess`

Figure 12: Training efficiency of `NeuralSolver` and Bansal et al. [6] on same-size tasks: we present the accuracy of the learned algorithms on extrapolating to problems with different dimensionality (columns). Each color represents a different training size, specific to each task (`Thin Maze`, `Chess`). In the dashed line we show the upper-bound on the performance.

## C.4    Ablation on Model Size of `NeuralSolver`

We conducted an additional evaluation of the extrapolation performance of our method, as done in Section 5.2, for different sizes of the LSTM's output channel in the different-size tasks. As expected, we can observe a reduction in the performance of our method across most tasks as we reduce the capacity of our LSTM. However, we point out that our smallest model still achieves competitive performances against Bansal et al. [6], as shown in Table 3.

## C.5    Ablation on the Recurrent Module for `NeuralSolver`

We compare the performance of our model with two different recurrent modules: LSTM (our default method), GRU [28] and LocRNN [29]. In Table 15, we present the results of this comparison, following the same procedure of Section 5.2. The results show that our LSTM module consistently outperforms the other alternatives. We also show the number of parameters and computational complexity in Table 16 and Table 17, respectively.

## C.6    Extended Ablation Results

In Table 18, we performed additional ablation on the extrapolation abilities when we removed the layer norm component of the recurrent neural network (No LN) and when we disabled the standard PyTorch dropout and Gal Dropout [27]. The results show that both components help improve extrapolation, especially in the `1S-Maze` task. We explored the use of a projection layer of the input like used in the Bansal et al. [6] baseline [6], and noticed a slight negative impact on the extrapolation.

## C.7    Extreme Extrapolation Results

We evaluate the performance of `NeuralSolver` and Bansal et al. [6] on extrapolating to very high-dimensional input observations. In particular, we evaluate the performance of the methods on

Table 11: Train classification accuracy of all models in the different-size tasks.

| Model | 1S-Maze | GoTo | Pong | Doorkey |
|---|---|---|---|---|
| NeuralSolver | 99.99 ±0.00 | 99.99 ±0.01 | 100.00 ±0.00 | 99.69 ±0.03 |
| Bansal et al. [6] | 100.00 ±0.00 | 99.99 ±0.01 | 100.00 ±0.00 | 99.97 ±0.02 |
| FeedForward | 100.00 ±0.00 | 100.00 ±0.00 | 100.00 ±0.00 | 99.85 ±0.05 |

Table 12: Validation classification accuracy of all models in the different-size tasks.

| Model | 1S-Maze | GoTo | Pong | Doorkey |
|---|---|---|---|---|
| NeuralSolver | 100.00 ±0.00 | 100.00 ±0.00 | 100.00 ±0.00 | 100.00 ±0.00 |
| Bansal et al. [6] | 100.00 ±0.00 | 100.00 ±0.00 | 100.00 ±0.00 | 100.00 ±0.00 |
| FeedForward | 100.00 ±0.00 | 100.00 ±0.00 | 100.00 ±0.00 | 99.94 ±0.04 |

extrapolating to $256 \times 256$ (Table 19) and $512 \times 512$ (Table 20) observations. Due to the large dimensionality of the tasks and the large number of iterations required to solve the tasks, we only consider 100 testing examples[2] The results show that only NeuralSolver is able to maintain perfect accuracy across all tasks. Moreover, our model does not suffer from overthinking, as shown in Figure 13.

## C.8   Hyperparameter Scan

In Fig. 14 we perform a small hyperparameter scan on the choosen default values in Table 9.

---

[2]Each test on the 1S-Maze task with observations of size $512 \times 512$ took on average 28 hours on a Nvidia 4090 GPU.

Table 13: Extrapolation accuracy for different values of $\alpha$ on the different-size tasks and in the `Thin-Maze` task.

| $\alpha$ | 1S-Maze | GoTo | Pong | Doorkey | Thin-Maze |
|------|---------|------|------|---------|-----------|
| 0.00 | $67.16 \pm 5.80$ | $62.98 \pm 12.21$ | $69.24 \pm 7.60$ | $93.81 \pm 6.84$ | $\mathbf{46.78} \pm \mathbf{37.49}$ |
| 0.01 | $63.47 \pm 12.73$ | $\mathbf{64.69} \pm \mathbf{9.23}$ | $\mathbf{71.97} \pm \mathbf{10.70}$ | $\mathbf{97.13} \pm \mathbf{1.46}$ | $42.76 \pm 37.56$ |
| 0.10 | $73.36 \pm 7.56$ | $62.09 \pm 13.38$ | $62.53 \pm 10.11$ | $93.44 \pm 4.71$ | $22.26 \pm 27.65$ |
| 0.50 | $\mathbf{74.14} \pm \mathbf{2.60}$ | $58.45 \pm 5.81$ | $68.13 \pm 13.27$ | $93.19 \pm 8.70$ | $0.22 \pm 0.17$ |
| 1.00 | $66.55 \pm 13.75$ | $60.52 \pm 11.75$ | $71.48 \pm 14.83$ | $94.71 \pm 2.96$ | $5.32 \pm 5.84$ |

Table 14: Extrapolation performance `NeuralSolver` in the proposed different-size tasks with different number of channels in the LSTM's output and hidden state (model width). Higher is better.

| Model Width | 1S-Maze | GoTo | Pong | Doorkey |
|-------------|---------|------|------|---------|
| 64 (Default) | $\mathbf{100.00} \pm \mathbf{0.00}$ | $\mathbf{100.00} \pm \mathbf{0.00}$ | $\mathbf{100.00} \pm \mathbf{0.00}$ | $\mathbf{100.00} \pm \mathbf{0.00}$ |
| 48 | $98.79 \pm 3.82$ | $99.94 \pm 0.20$ | $100.00 \pm 0.00$ | $99.76 \pm 0.63$ |
| 32 | $78.75 \pm 3.82$ | $96.56 \pm 10.13$ | $100.00 \pm 0.00$ | $98.65 \pm 2.89$ |
| 24 | $80.05 \pm 3.31$ | $96.96 \pm 7.91$ | $100.00 \pm 0.00$ | $91.21 \pm 17.69$ |

Table 15: Extrapolation accuracy of `NeuralSolver` with different recurrent modules in the proposed different-size tasks. Higher is better.

| Model | 1S-Maze | GoTo | Pong | Doorkey |
|-------|---------|------|------|---------|
| LSTM (Default) | $\mathbf{100.00} \pm \mathbf{0.00}$ | $\mathbf{100.00} \pm \mathbf{0.00}$ | $\mathbf{100.00} \pm \mathbf{0.00}$ | $\mathbf{100.00} \pm \mathbf{0.00}$ |
| GRU [28] | $83.24 \pm 10.09$ | $99.94 \pm 0.13$ | $\mathbf{100.00} \pm \mathbf{0.00}$ | $95.85 \pm 4.34$ |
| LocRNN [29] | $87.65 \pm 9.13$ | $82.56 \pm 17.73$ | $94.08 \pm 8.81$ | $86.02 \pm 11.75$ |

Table 16: Total parameter count of `NeuralSolver` with different recurrent modules in the proposed different-size tasks, in the scale of millions of parameters. Lower is better.

| Model | 1S-Maze | GoTo | Pong | Doorkey |
|-------|---------|------|------|---------|
| LSTM (Default) | 0.231 | 0.231 | 0.230 | 0.231 |
| GRU [28] | **0.192** | **0.192** | **0.192** | **0.192** |
| LocRNN [29] | 0.236 | 0.236 | 0.236 | 0.236 |

Table 17: Computational complexity (in gigaMACs) of `NeuralSolver` with different recurrent modules in the proposed different-size tasks. Lower is better.

| Model | 1S-Maze | GoTo | Pong | Doorkey |
|-------|---------|------|------|---------|
| LSTM (Default) | 4.12 | 9.76 | 9.76 | 9.76 |
| GRU [28] | **3.19** | **7.55** | **7.54** | **7.55** |
| LocRNN [29] | 4.33 | 10.25 | 10.25 | 10.25 |

Table 18: Extended extrapolation accuracy of different ablated versions of `NeuralSolver` in the proposed different-size tasks. Higher is better.

| Model | 1S-Maze | GoTo | Pong | DoorKey |
|---|---|---|---|---|
| `NeuralSolver` | **100.00** ±0.00 | **100.00** ±0.00 | **100.00** ±0.00 | **100.00** ±0.00 |
| No LN | 75.53 ± 0.87 | 99.55 ±1.49 | 100.00 ±0.00 | 97.11 ±1.99 |
| No Dropout | 90.61 ±11.23 | 98.48 ±3.05 | 100.00 ±0.00 | 99.65 ±0.89 |
| With Projection | 97.24 ± 8.25 | 99.93 ±0.21 | 100.00 ±0.00 | 99.45 ±1.02 |

Table 19: Extrapolation accuracy on different-size tasks considering $256 \times 256$ observation size.

| Model | 1S-Maze | GoTo | Pong | Doorkey |
|---|---|---|---|---|
| `NeuralSolver` | **100.00** ±0.00 | **100.00** ± 0.00 | **100.00** ±0.00 | **100.00** ±0.00 |
| Bansal et al. [6] | 77.20 ±2.48 | 60.33 ±12.38 | 65.00 ±9.01 | 100.00 ±0.00 |

Table 20: Extrapolation accuracy on different-size tasks considering $512 \times 512$ observation size.

| Model | 1S-Maze | GoTo | Pong | Doorkey |
|---|---|---|---|---|
| `NeuralSolver` | **100.00** ±0.00 | **100.00** ± 0.00 | **100.00** ±0.00 | **100.00** ±0.00 |
| Bansal et al. [6] | 79.20 ±5.81 | 57.22 ±10.15 | 63.60 ±2.42 | 99.71 ±0.45 |

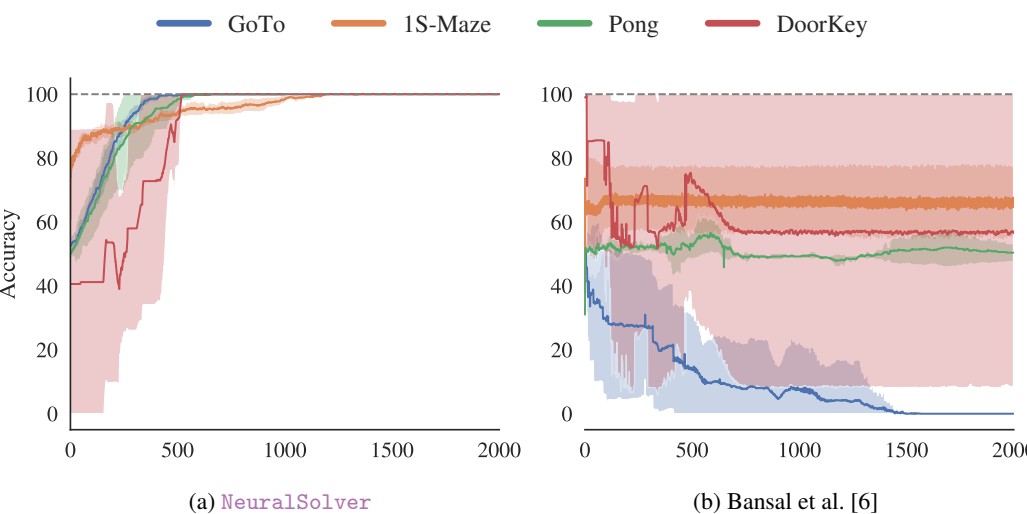

(a) `NeuralSolver`  (b) Bansal et al. [6]

Figure 13: Accuracy results of extrapolation to large observation sizes (512×512). All results are averaged over 10 randomly-selected seeds. In the `1S-Maze` environment, the real number of iterations is 10x larger. Higher is better.

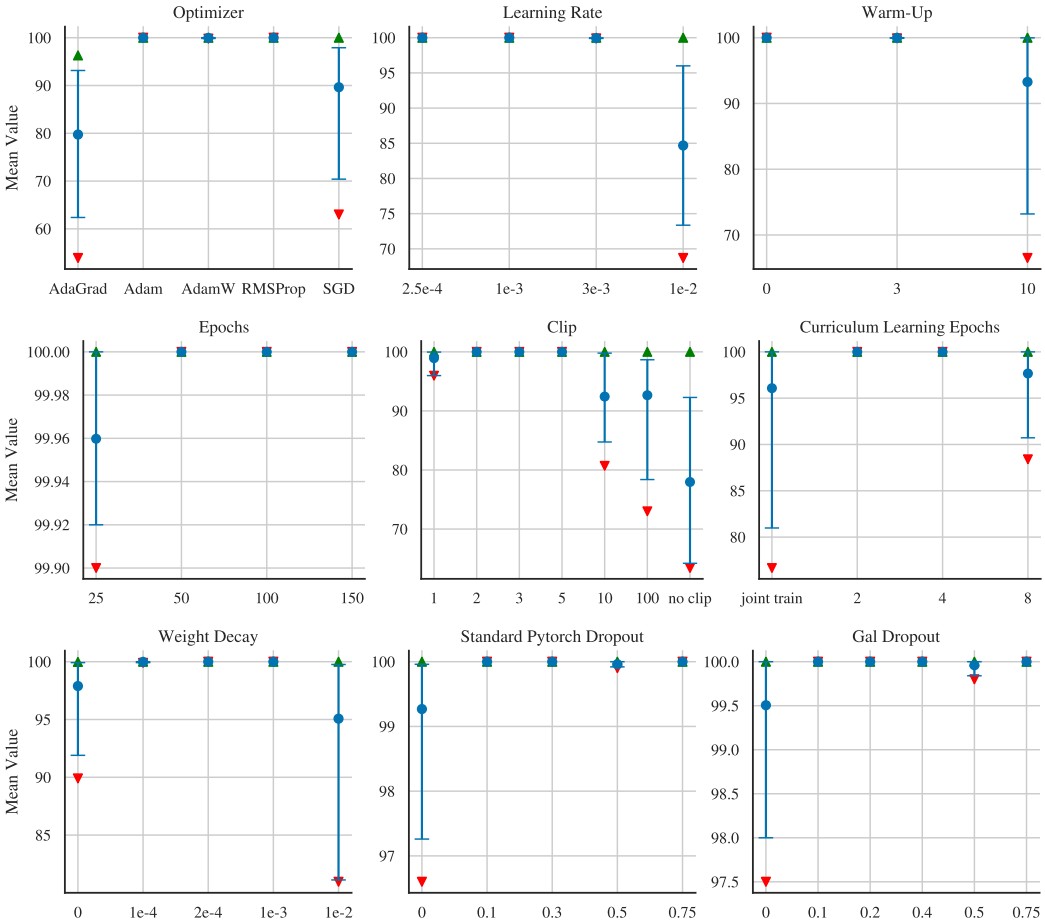

Figure 14: Hyperparameter scan of `NeuralSolver` on the `GoTo` task. Each plot makes a single change from the hyperparameters in Table 9. The blue whiskers represent the confidence intervals at 95%. The green and red arrows represent the maximum and minimum values of the bootstrap distribution, respectively.

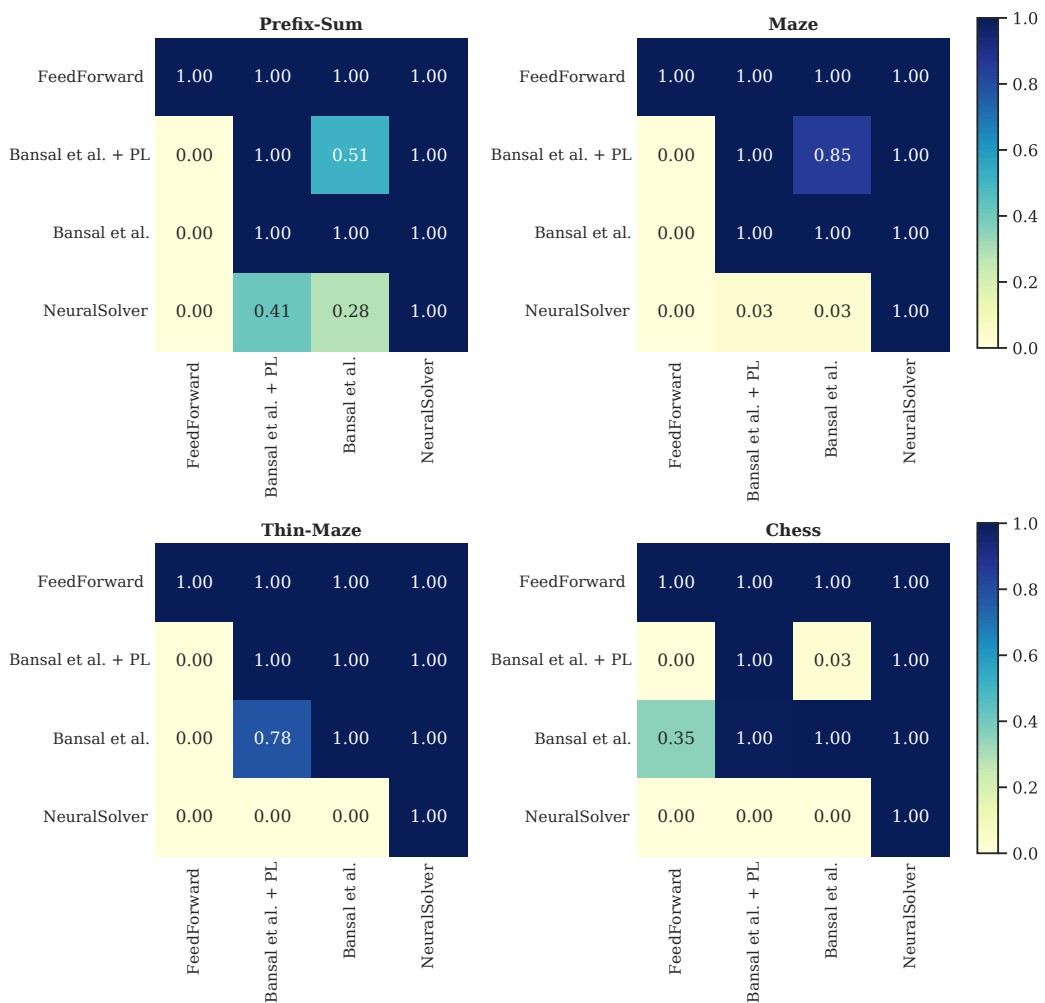

Figure 15: Almost Stochastic Order scores of the same-size task results presented in Section 5.1. ASO scores are expressed in $\epsilon_{\min}$, with a significance level $\alpha = 0.05$ that is adjusted accordingly by using the Bonferroni correction [17]. Read from row to column: e.g., `NeuralSolver` (row) is almost stochastically dominant over Bansal et al. [6] (column) in the `Prefix-Sum` task with $\epsilon_{\min}$ of 0.28.

# D  Statistical Significance

The Almost Stochastic Order test (ASO; [15, 16]) was recently introduced to assess statistical significance in Deep Neural Networks across multiple runs. The ASO test measures the presence of a stochastic order between two models or algorithms based on their respective sets of evaluation scores. Using the individual scores of algorithm A and B across various random seeds, the method calculates a test-specific value ($\epsilon_{\min}$), representing how far algorithm A is from being significantly superior to algorithm B. The lower the value of $\epsilon_{\min}$, the more likely it is that A is better than B. For $\epsilon_{\min} < 0.5$, A is said to be *almost stochastically dominant* over B in most cases than vice versa, and thus A could be considered superior to B, although with less confidence. We claim that A is *stochastically dominant* over B with a predefined significance level when $\epsilon_{\min} = 0.0$. We use the implementation by [30] to perform these statistical tests. We present the comparison of all pairs of models using ASO with a confidence level of $\alpha = 0.05$ (before adjusting for all pair-wise comparisons using the Bonferroni correction [17]).

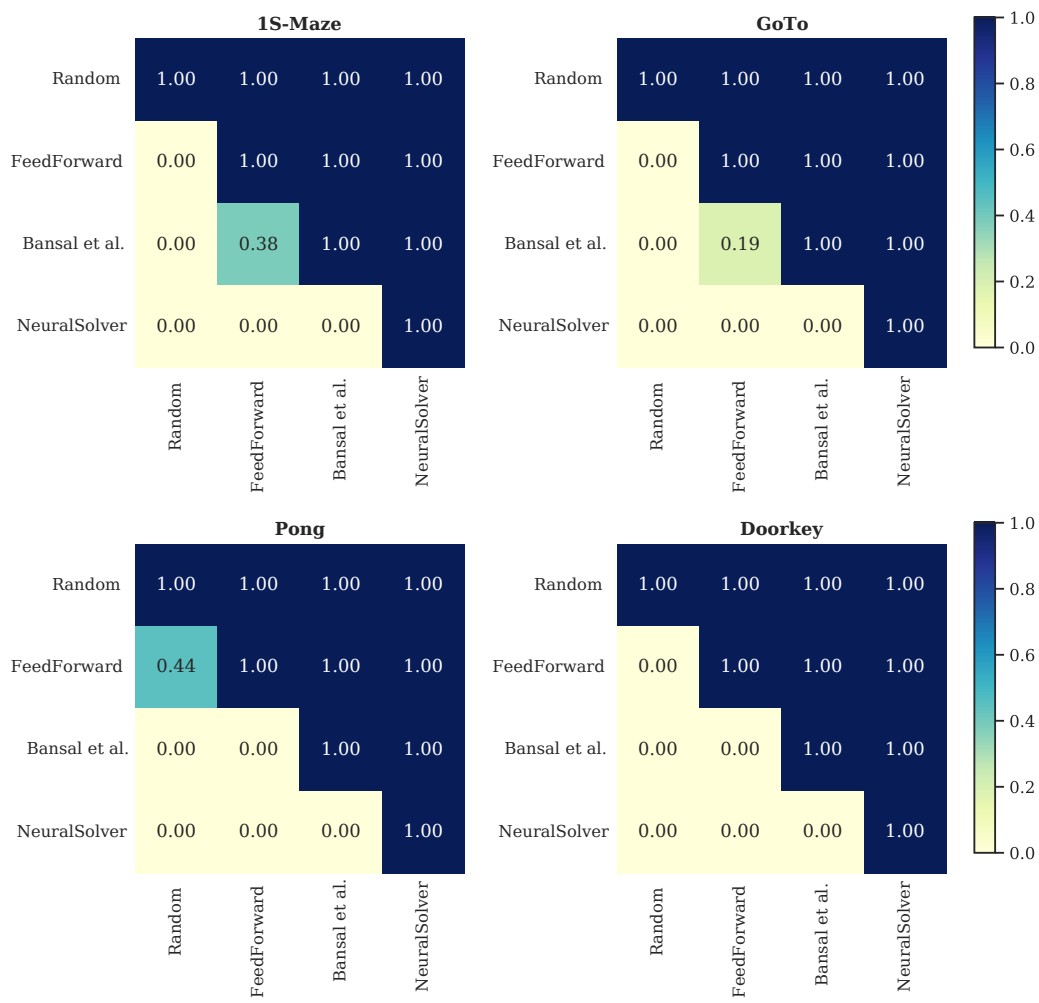

Figure 16: Almost Stochastic Order scores of the different-size task results presented in Section 5.2. ASO scores are expressed in $\epsilon_{\min}$, with a significance level $\alpha = 0.05$ that is adjusted accordingly by using the Bonferroni correction [17]. Read from row to column: e.g., `NeuralSolver` (row) is stochastically dominant over Bansal et al. [6] (column) in the `1S-Maze` task with $\epsilon_{\min}$ of 0.00.

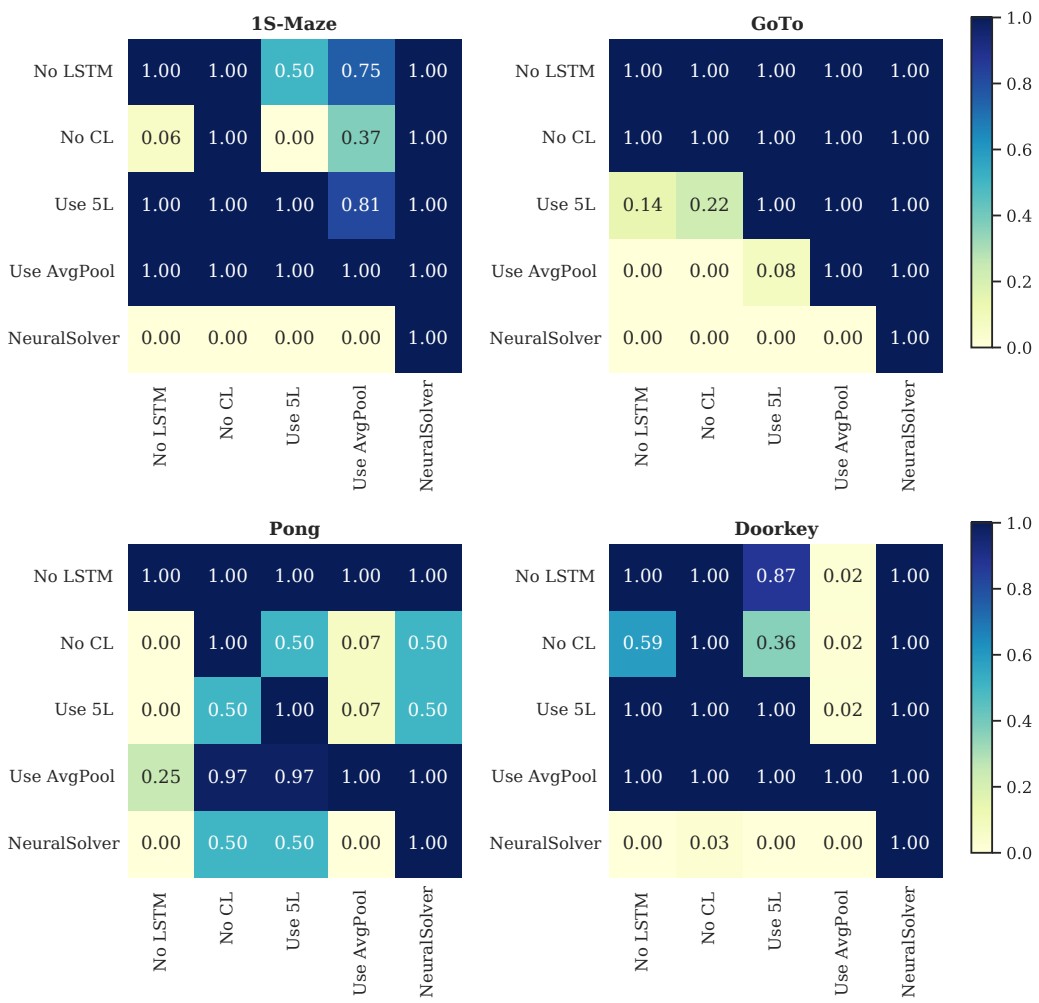

Figure 17: Almost Stochastic Order scores of the ablation study on the different-size tasks presented in Section 5.3. ASO scores are expressed in $\epsilon_{\min}$, with a significance level $\alpha = 0.05$ that is adjusted accordingly by using the Bonferroni correction [17]. Read from row to column: e.g., `NeuralSolver` (row) is stochastically dominant over No LSTM (column) in the `1S-Maze` task with $\epsilon_{\min}$ of 0.00.

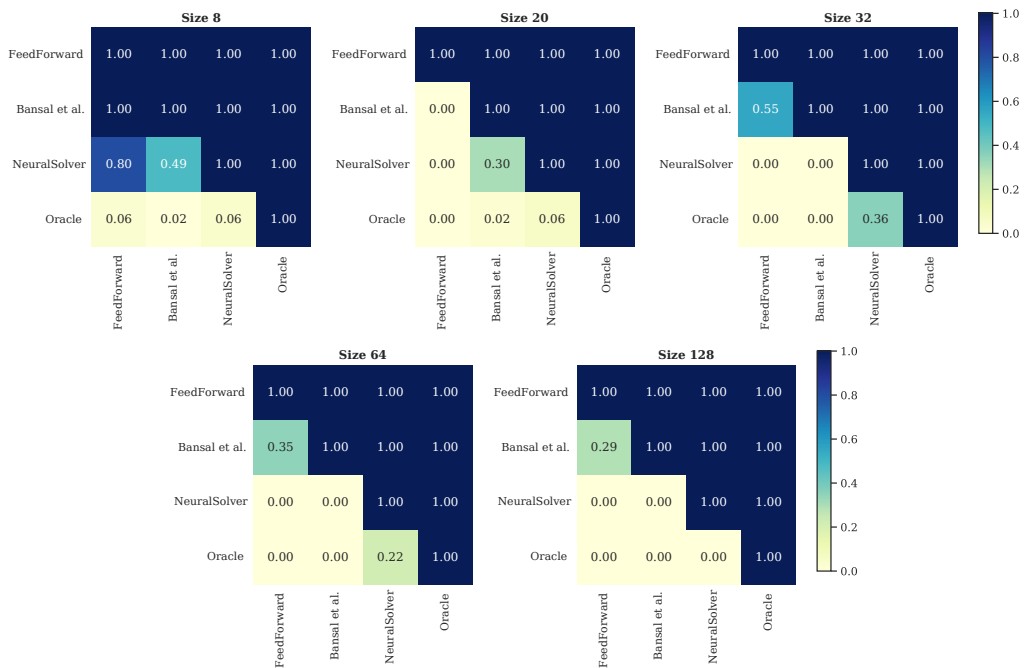

Figure 18: Almost Stochastic Order scores of the `Doorkey` sequential decision-making task results presented in Section 5.4. ASO scores are expressed in $\epsilon_{\min}$, with a significance level $\alpha = 0.05$ that is adjusted accordingly by using the Bonferroni correction [17]. Read from row to column: e.g., `NeuralSolver` (row) is stochastically dominant over Bansal et al. [6] (column) starting at scenarios with observations of size 32 with an $\epsilon_{\min}$ of 0.00.

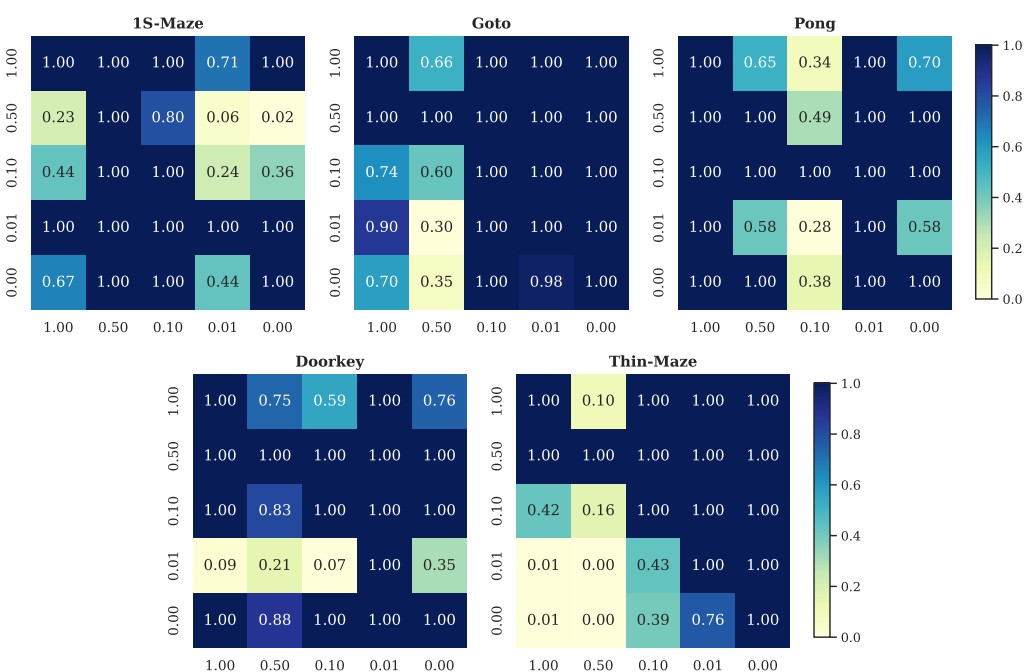

Figure 19: Almost Stochastic Order scores of the fine tuning progressive loss results presented in Appendix C.3. ASO scores are expressed in $\epsilon_{\min}$, with a significance level $\alpha = 0.05$ that is adjusted accordingly by using the Bonferroni correction [17]. Read from row to column: e.g., a progressive loss alpha value of 0.5 (row) is almost stochastically dominant over the value of 1. (column) in the `1S-Maze` task with $\epsilon_{\min}$ of 0.23.

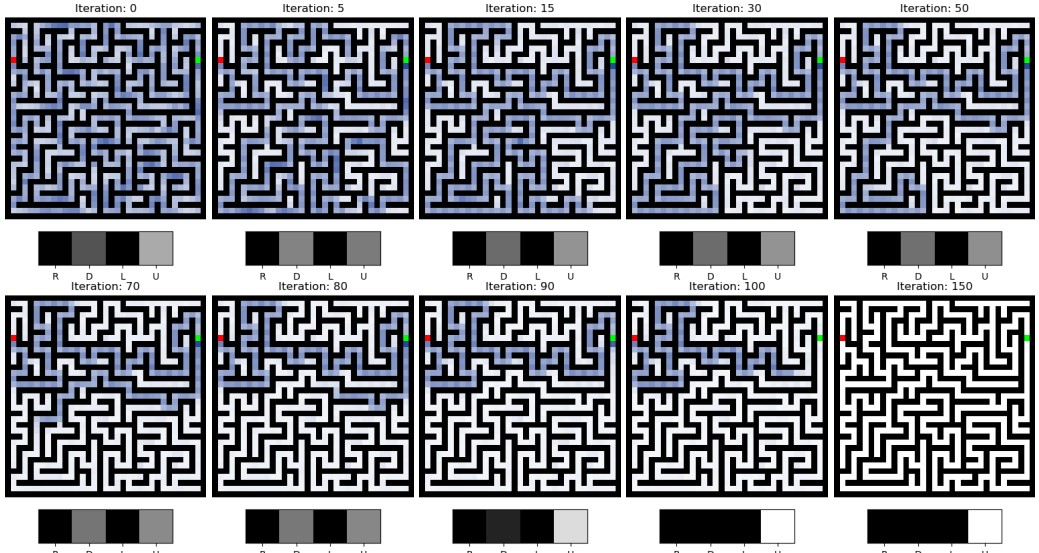

Figure 20: Propagation of information in `NeuralSolver` for the `1S-Maze` task: Top: we highlight the difference between the current iteration and last iteration (not represented) of the internal state of the recurrent module of our model in a $35 \times 35$ environment. The white pixels represent the pixel positions of the recurrent state that converged to a fixed final state. Larger differences are represented in deep blue color. Bottom: the predicted action probabilities by the model at different iterations, where the agent can move right (R), down (D), left (L), or up (U).

# E    Learned Algorithms with `NeuralSolver`

Similar to Section 3.2, we show visualizations of the recurrent space of `NeuralSolver` while solving the different-size tasks. More specifically, we visualize the difference between the current recurrent state of the LSTM and the last recurrent state, for each iteration. The following figures show two plots per iteration: the one on top shows the difference between the recurrent states in a certain color superimposed over the input image, while the one below shows the predicted action probabilities outputted by the model.

With these visualizations we notice how the information propagates through the input problems, giving a hint of the algorithm that the model is performing. An interesting observation across these visualizations is how the final prediction of the model only converges to the correct action after the recurrent iterations have converged to a fixed state around the player position.

## E.1    `1S-Maze`

By looking at how the information propagates through the maze paths in Figure 20 and 21, the model appears to learn a parallel algorithm that starts with the dead ends until finding the optimal path between the player (green) and the goal (red). At iteration #90 in Figure 20, the visible difference between the current and last recurrent states actually represents the optimal path between the player and the goal. The algorithm converges to the optimal task after the difference of the state converges near the player. Figure 21 shows that finding the optimal path is not necessary to predict the next action.

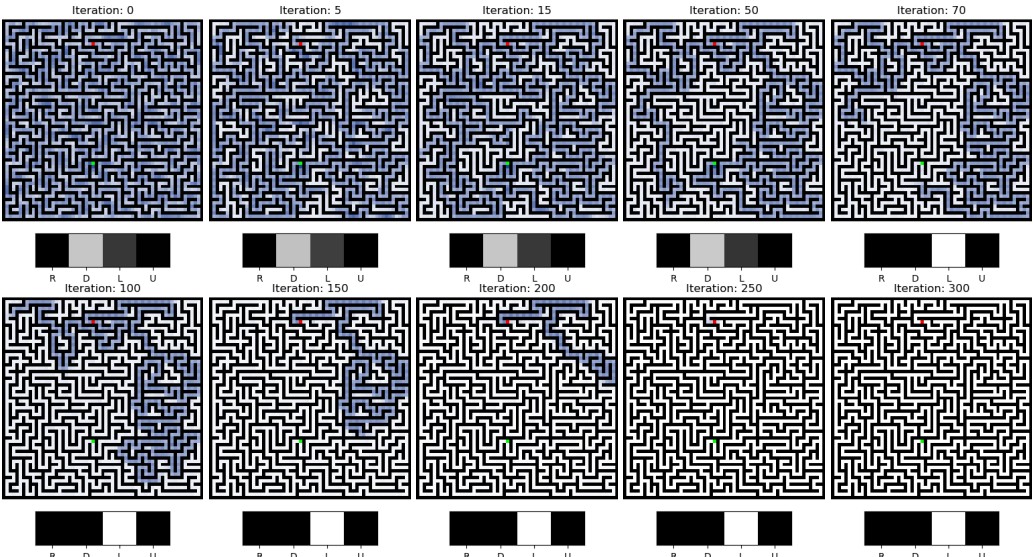

Figure 21: Propagation of information in `NeuralSolver` for the `1S-Maze` task: Top: we highlight the difference between the current iteration and last iteration (not represented) of the internal state of the recurrent module of our model in a $61 \times 61$ environment. The white pixels represent the pixel positions of the recurrent state that converged to a fixed final state. Larger differences are represented in deep blue color. Bottom: the predicted action probabilities by the model at different iterations, where the agent can move right (R), down (D), left (L), or up (U).

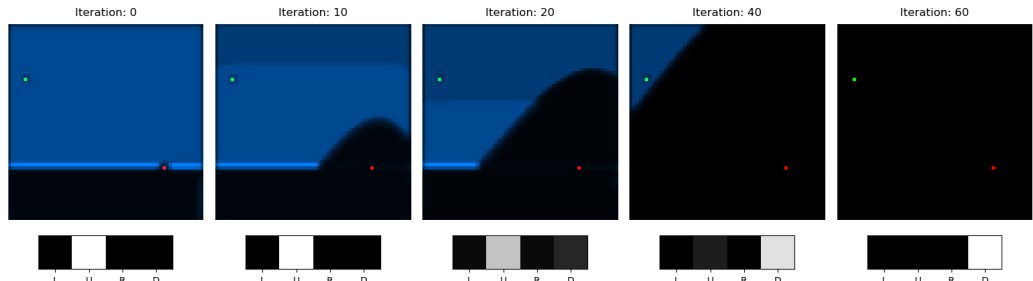

Figure 22: Propagation of information in `NeuralSolver` for the `GoTo` task: Top: we highlight the difference between the current iteration and last iteration (not represented) of the internal state of the recurrent module of our model in a $64 \times 64$ environment. The black pixels represent the pixel positions of the recurrent state that converged to a fixed final state. Larger differences are represented in deep blue color. Bottom: the predicted action probabilities by the model at different iterations, where the agent can move right (R), down (D), left (L), or up (U).

## E.2 `GoTo`

We highlight three examples of possible player and goal positions that influence the final predicted action. Namely, if the player (green) is above the goal (red), below the goal, or at the same level in height.

In Figure 22, we can notice that the model chooses by default the up action, while across the iterations a signal is sent from the goal position to the positions above, propagating in a circular/oval shape. Thus the model appears to learn an algorithm that attempts to signal the player if it should go down. Similarly, in Figure 23 the recurrent state does not change near the player, as such the predicted action remains the same.

In Figure 24 we see a case in which the player is on the left of the goal. In this setting, the horizontal line with a slightly different contrast ranging from the goal position appears to communicate across that line the player should go left or right to reach the goal.

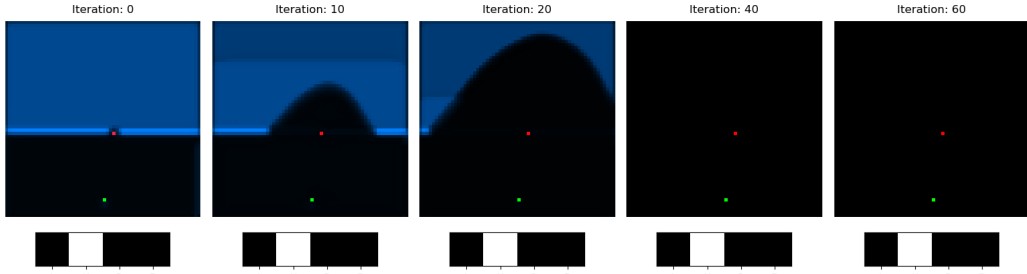

Figure 23: Propagation of information in `NeuralSolver` for the `GoTo` task: Top: we highlight the difference between the current iteration and last iteration (not represented) of the internal state of the recurrent module of our model in a $64 \times 64$ environment. The black pixels represent the pixel positions of the recurrent state that converged to a fixed final state. Larger differences are represented in deep blue color. Bottom: the predicted action probabilities by the model at different iterations, where the agent can move right (R), down (D), left (L), or up (U).

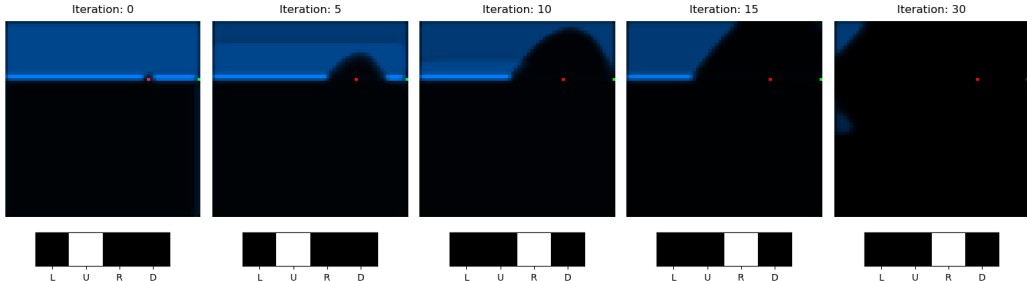

Figure 24: Propagation of information in `NeuralSolver` for the `GoTo` task: Top: we highlight the difference between the current iteration and last iteration (not represented) of the internal state of the recurrent module of our model in a $64 \times 64$ environment. The black pixels represent the pixel positions of the recurrent state that converged to a fixed final state. Larger differences are represented in deep blue color. Bottom: the predicted action probabilities by the model at different iterations, where the agent can move right (R), down (D), left (L), or up (U).

### E.3 `Pong`

We present two examples our learned algorithms solving the simplified game of `Pong`: one when the ball is on the left of the paddle, and the other when on the right. We can notice that the model learns a similar algorithm to the model in the `GoTo` task. In both cases the model starts by predicting that the paddle should move right. In Figure 25, the model prediction changes once the information propagation reaches the paddle, changing it to the left action. In Figure 26, we see that this propagation of information does not reach the paddle, and as such the action remains unchanged.

We can also see a line with a slightly different contrast leaving the center of the ball, thus signaling that the model should instead predict the stay action. This could indicate that different contrasts might represent different algorithms running simultaneously.

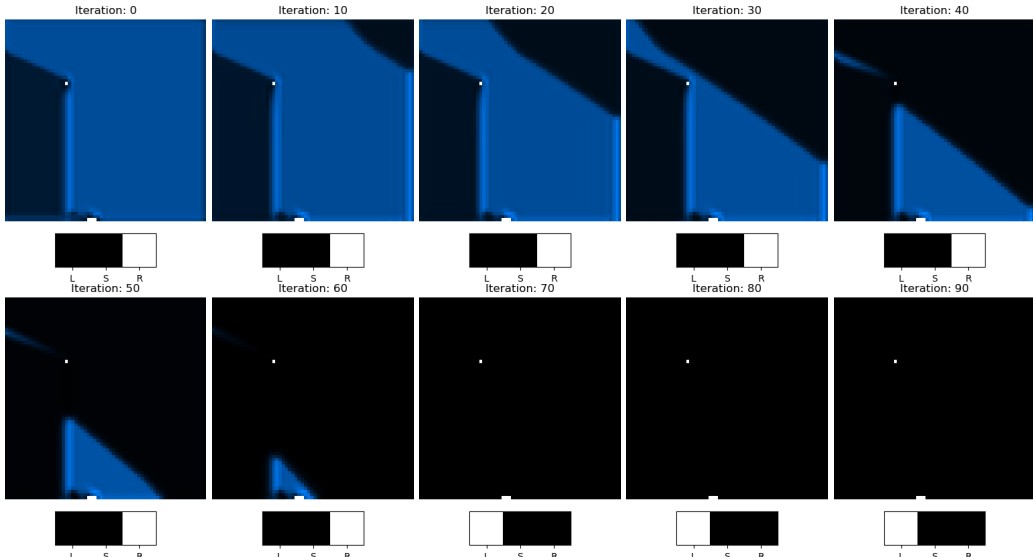

Figure 25: Propagation of information in `NeuralSolver` for the `Pong` task: Top: we highlight the difference between the current iteration and last iteration (not represented) of the internal state of the recurrent module of our model in a $64 \times 64$ environment. The black pixels represent the pixel positions of the recurrent state that converged to a fixed final state. Larger differences are represented in deep blue color. Bottom: the predicted action probabilities by the model at different iterations, where the agent can move left (L), stay (S) and right (R).

## E.4 `Doorkey`

We provide two examples of learned algorithms in `Doorkey`: in Figure 27 we show a case when the player is almost reaching the goal, and in Figure 28 a case when it already captured the key and is reaching for the door.

These visualizations are harder to interpret than the other ones. This phenomenon might occur since, to solve this task, the agent needs to follow a complex sequence of tasks, thus requiring multiple algorithms to find where the positions of the objects are and which action to do next. We can notice this in Figure 28, where there is a line that shoots vertically from the door position that is not present in Figure 27. This shows that for more complex tasks, better visualization and interpretation techniques of the recurrent states are required.

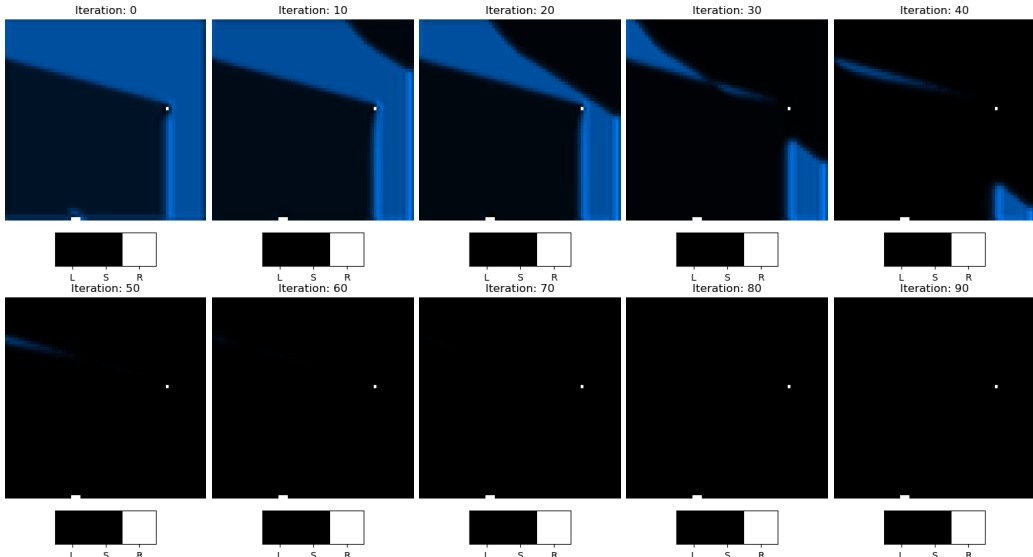

Figure 26: Propagation of information in `NeuralSolver` for the `Pong` task: Top: we highlight the difference between the current iteration and last iteration (not represented) of the internal state of the recurrent module of our model in a $64 \times 64$ environment. The black pixels represent the pixel positions of the recurrent state that converged to a fixed final state. Larger differences are represented in deep blue color. Bottom: the predicted action probabilities by the model at different iterations, where the agent can move left (L), stay (S) and right (R).

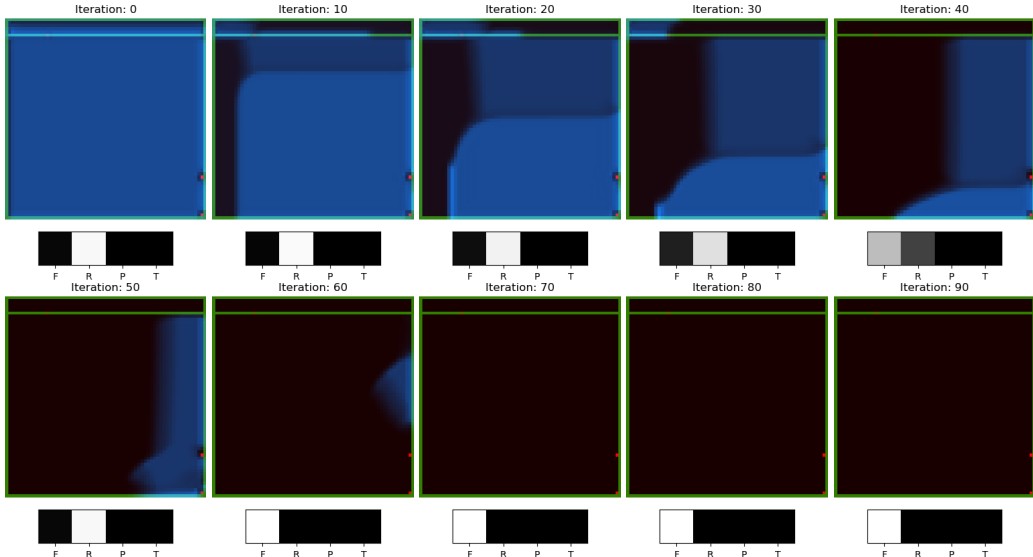

Figure 27: Propagation of information in `NeuralSolver` for the `DoorKey` task: Top: we highlight the difference between the current iteration and last iteration (not represented) of the internal state of the recurrent module of our model in a $64 \times 64$ environment. The black pixels represent the pixel positions of the recurrent state that converged to a fixed final state. Larger differences are represented in deep blue color. Bottom: the predicted action probabilities by the model at different iterations, namely Forward (L), Rotate Right (R), Pickup (P), and Toggle (T).

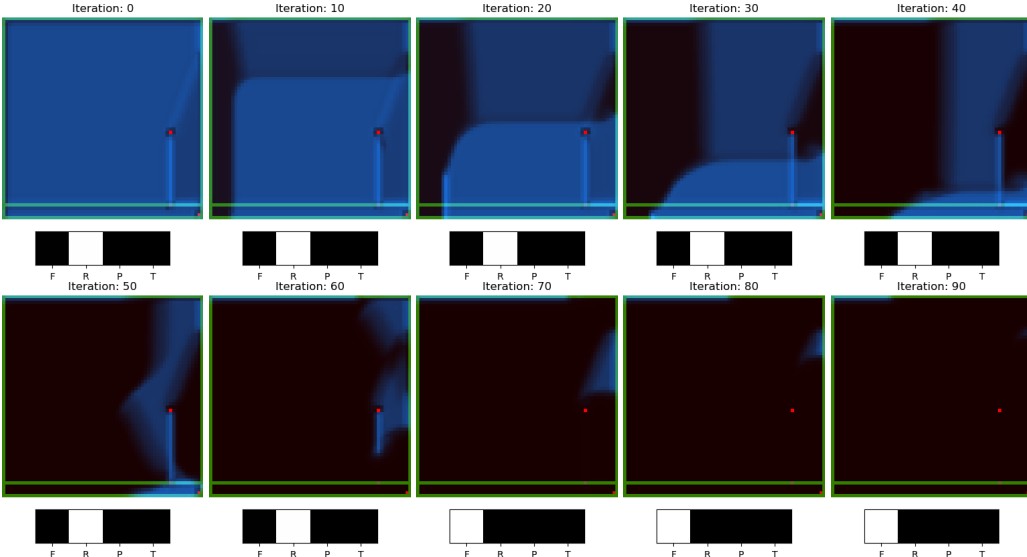

Figure 28: Propagation of information in `NeuralSolver` for the `DoorKey` task: Top: we highlight the difference between the current iteration and last iteration (not represented) of the internal state of the recurrent module of our model in a $64 \times 64$ environment. The black pixels represent the pixel positions of the recurrent state that converged to a fixed final state. Larger differences are represented in deep blue color. Bottom: the predicted action probabilities by the model at different iterations, namely Forward (L), Rotate Right (R), Pickup (P), and Toggle (T).

# F  Additional Example Trajectories

In Figure 29 we present additional trajectories of agents that execute policies learned using
`NeuralSolver` and other baselines, following the same procedure as in Section 5.4.

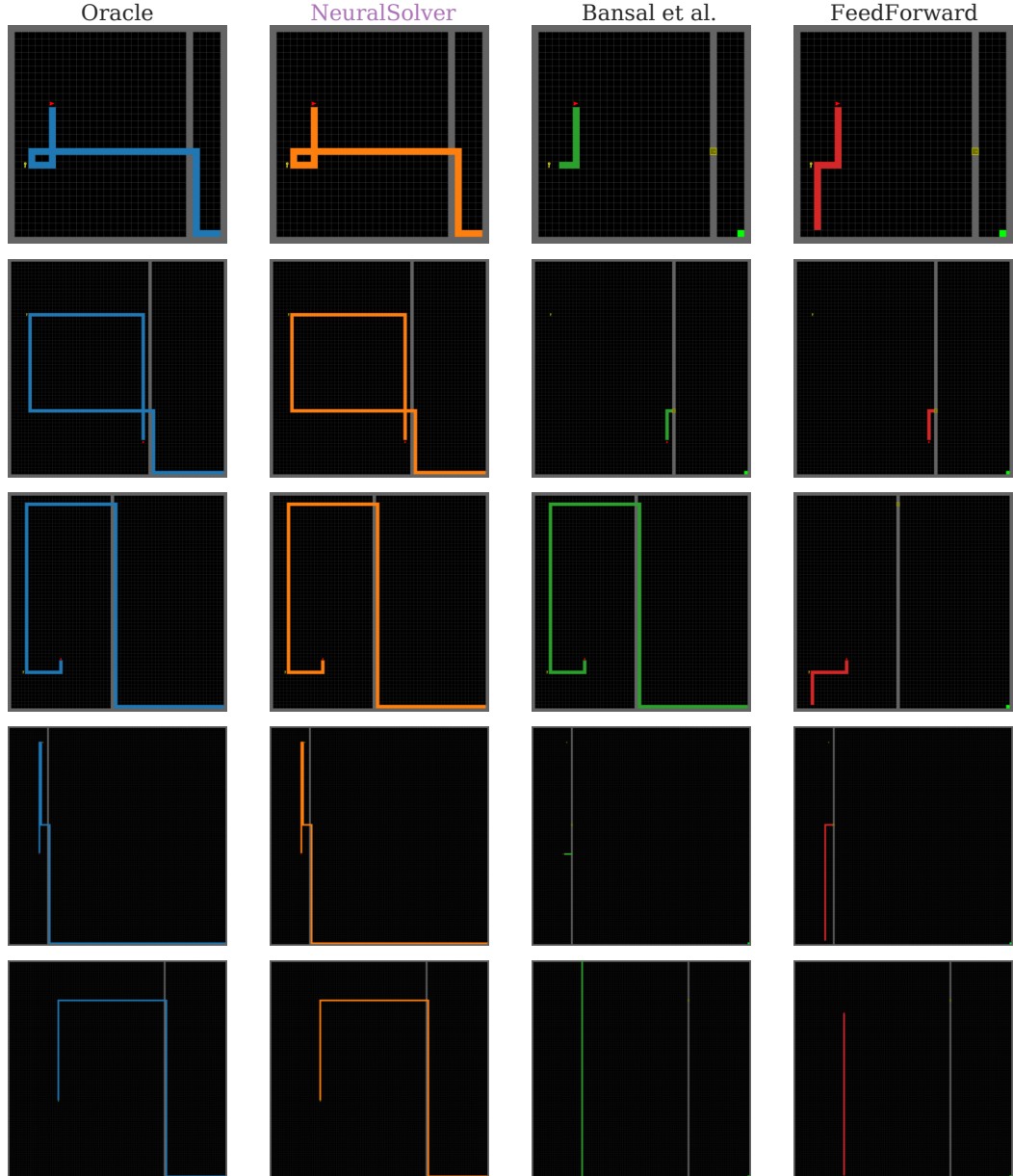

Figure 29: Visualization of the trajectories of `NeuralSolver`, Bansal et al. [6], and FeedForward
models against the Oracle trajectory. The first row is a task of size $32 \times 32$, second and third of
$64 \times 64$, and fourth and fifth of $128 \times 128$.

# G   Examples of tasks with observations of size $512 \times 512$

We present additional visualizations of examples of tasks with large observations ($512 \times 512$).

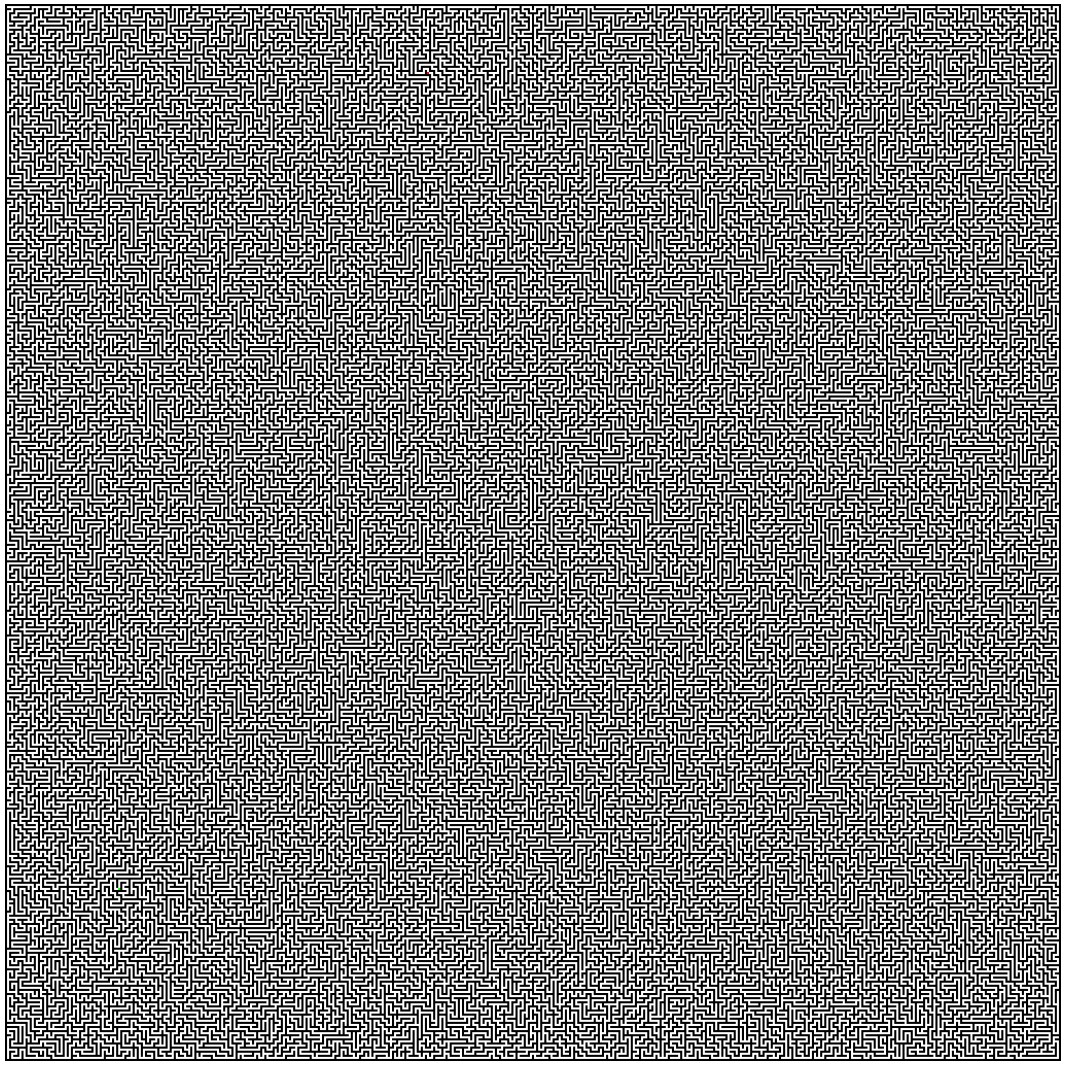

Figure 30: Example of a `1S-Maze` task with $512 \times 512$ observations.

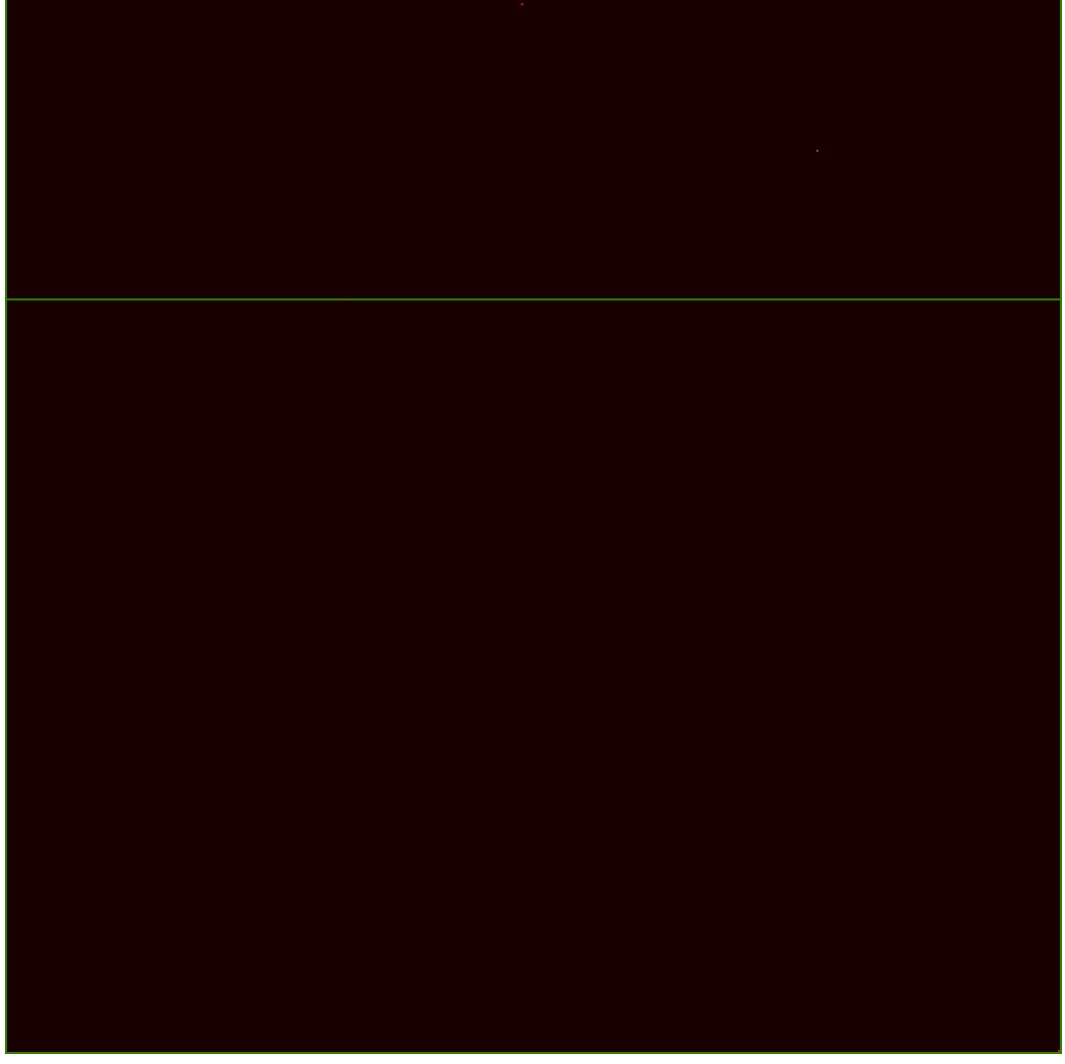

Figure 31: Example of a DoorKey task with $512 \times 512$ observations.

