# OpenReview forum: "NeuralSolver: Learning Algorithms For Consistent and Efficient Extrapolation Across General Tasks"
_NeurIPS.cc/2024/Conference — NeurIPS 2024 poster_

### Official Review · Reviewer_dmPv · 2024-07-01

**Soundness:** 2
**Presentation:** 2
**Contribution:** 3
**Rating:** 5
**Confidence:** 2

**Summary:**

The paper proposes a method called "NeuralThink", which is designed to improve the same-size task generalization and different-size task extrapolation performances. The proposed algorithm composes of three components: (1) A recurrent module utilizing a LSTM network to process inputs of different scale, (2) A processing module serving as the aggregation layer, and (3) A curriculum learning training scheme  that gradually increases the dimensionality of the observation. Experiments on an algorithm learning benchmark showed that the proposed approach surpassed the baselines being compared, especially in terms of the different-size task extrapolation performance. It is also demonstrated that the proposed approach has a higher efficiency both in terms of training size and parameter size.

**Strengths:**

- The paper addresses a meaningful problem space: Algorithm learning and extrapolation ability. The proposed method achieves oracle-level performance on different-size tasks, showcasing the improved generalization and extrapolation power.
- The proposed approach outperforms the selected baselines on an algorithm learning benchmark, proving the effectiveness to a certain extent. Further experiments also indicates improved training size and parameter efficiency. The ablation study demonstrates that all components in the proposed approach contribute positively to the final result.
- The intro section is well-written. Figure 1-3 are well-illustrated.

**Weaknesses:**

- It's not immediately clear why the proposed approach achieves improved performance above prior arts. Explanation on (1) the fundamental difference compared to the previous approach, and (2) why this should work better would be helpful.
- The result being presented in the experiment section is a bit counter-intuitive. Table 3 shows that the proposed approach achieves oracle-level performance on different-size task extrapolation, yet Table 1 shows that the performance on same-size tasks is imperfect. Intuitively one would imagine different-size tasks to be more challenging for the algorithm, yet the result suggests otherwise. An analysis on the intuition behind would be helpful. Besides, more case studies to compare the performance gap between the proposed approach v.s. prior baselines would help solidify the understanding.
- The number of baselines being compared to is limited. It would be helpful to know how does the proposed approach compare to other recently published methods, e.g. [1], or if they are not directly comparable, what is the reason behind.
- nit: The presentation of some figures need to be improved for better clarity. e.g. Figure 5 it's not immediately clear what does the notion "x/y" mean in the legend of the bar chart.

[1] Adaptive recurrent vision performs zero-shot computation scaling to unseen difficulty levels. NeurIPS 2023.

**Questions:**

See above.

---

> ### Author Rebuttal · Authors · 2024-08-06
>
> We thank the reviewer for their insightful comments and the time spent reviewing our paper. We now address the weaknesses (W) identified by the reviewer:
>
> + **(W1) Differences between NeuralThink and the previous work/why NeuralThink achieves improved performance.** We would like to highlight the following fundamental differences between our work and previous work (and why):
>   1. **Recurrent design:** Previous work uses the simplest form of recurrence by using a ResNet with weight sharing across residual blocks. In our work, we started by evaluating the extrapolation capabilities of DeepThink and found its performance to be lacking in different-sized tasks (as we show in Table 3). To overcome this gap, we explored other more advanced recurrent designs for algorithmic extrapolation, in particular LSTM-based convolutional designs. Our results show that this choice of recurrent design results in better performance of the method, supporting prior work \[3\].
>   1. **Aggregation layer in different size tasks:** Previous DeepThinking methods do not work in this class of problems, as their models only support same size input and output problems. To overcome this issue, we introduced the use of a pooling layer in the design of our model.
>   1. **Curriculum-based training scheme:** The previous differences are insufficient to allow consistent extrapolation in different-size tasks. To the best of our knowledge, we are the first work that introduces a curriculum-based training scheme for DeepThinking architectures. In Table 4 we highlight the role of our curriculum-based training scheme in the overall performance of the method.
>
>    We provide in Appendix B.6. of the original paper an extended discussion on the differences between our model and DeepThink. We have clarified the existence of this comparison in line 129 of the updated version of the paper.
>
> + **(W2) Imperfect results in same-sized tasks conflict with oracle-level performance on different-size task extrapolation:** We would like to point out that, while the result for Thin-Maze is not at oracle-level accuracy, the results are still inside of the experimental uncertainty of the other results. On the other hand, the Chess task is fundamentally different from the other tasks, since it is not a spatial extrapolation task but a symbolic extrapolation level, proving to be a harder task for our method. However, despite not achieving oracle-level performance, even in this challenging task our model is able to outperform the previous state-of-the-art.
>
> + **(W2,1) The number of used tasks is small:** We would like to point out that not only are we using all the benchmarks proposed by \[2\] to evaluate DeepThinking models, but we also contribute a novel set of different-sized tasks, due to the absence of an existing adequate benchmark for this type of tasks. Furthermore, we are the first work to explore sequential decision-making scenarios for the application of DeepThinking models.
>
> + **(W3) Limited Baselines/AdRNNs**: We have updated our related section with \[1\]. We have compared directly and indirectly the performance of AdRNNs to NeuralThink.
>   + **AdRNNs have a worse performance than DeepThink (our baseline) on same-sized tasks.** \[1\] uses the same training methodology as NeuralThink on the same-sized Maze tasks (train on mazes of size 24x24 pixels) but test on much smaller mazes (test on mazes of size 44x44 and 56x56 pixels) than in our paper. The best method presented in \[1\] (LocRNN) achieves 50% (best performance) in test scenarios that are half the size of the ones presented in our work: DeepThink achieves 91% test accuracy on mazes of size 124x124. As such, DeepThink is a stronger baseline than AdRNNs for algorithmic extrapolation and, thus, we outperform them as well.
>   + **AdRNNs have a worse performance than NeuralThink on different-sized tasks.** Despite the fact that no public code implementation of \[1\] is available, we implemented the proposed LocRNN without the learnable iteration-halting mechanism (as we always select the best performing iteration for the results) and compared its performance against the ConvLSTM used in NeuralThink on the different-sized tasks. The results are shown in the below table, which highlights that LocRNN has a worse performance than our method, while having more parameters than NeuralThink and being more computationally intensive to execute. We added this additional comparison to Appendix C.5.
>
> | Extrapolation accuracy (%) | 1S-Maze | GoTo | Pong | Doorkey |
> | :---- | :---- | :---- | :---- | :---- |
> | NeuralThink | 100.00 \+- 0.00  | 100.00 \+- 0.00  | 100.00 \+- 0.00  | 100.00 \+- 0.00  |
> | LocRNN \[1\] | 87.65 \+- 9.13 | 82.56 \+- 17.73 | 94.08 \+- 8.81 | 86.02 \+- 11.75 |
>
>
> | Number of parameters (in Millions) | 1S-Maze | GoTo | Pong | Doorkey |
> | :---- | :---- | :---- | :---- | :---- |
> | NeuralThink | 0.231 | 0.231 | 0.230 | 0.231 |
> | LocRNN \[1\] | 0.236 | 0.236 | 0.236 | 0.236 |
>
> | Computational complexity (in gigaMACs) | 1S-Maze | GoTo | Pong | Doorkey |
> | :---- | :---- | :---- | :---- | :---- |
> | NeuralThink | 4.12 | 9.76 | 9.76 | 9.76 |
> | LocRNN \[1\] | 4.33 | 10.25 | 10.25 | 10.25 |
>
> + **(W4) “x/y” color labels in Figure 5 are not clear.**
>   We changed the labels of the colors with very small, small, medium and large, and moved details of the training sizes used for each task to the Appendix.
>
> \[1\] Adaptive recurrent vision performs zero-shot computation scaling to unseen difficulty levels. NeurIPS 2023\.
> \[2\] Bansal, Arpit, et al. "End-to-end algorithm synthesis with recurrent networks: Extrapolation without overthinking." Advances in Neural Information Processing Systems 35 (2022)
> \[3\] Eric Price, Wojciech Zaremba, and Ilya Sutskever. Extensions and limitations of the neural GPU. CoRR, 337 abs/1611.00736, 2016\. doi: 10.48550/arxiv.1611.00736.

---

> ### Comment · Reviewer_dmPv · 2024-08-14
>
> Thanks sincerely for the detailed response from the authors. Raising the score to 5 since the concerns regarding lack of baseline / comparison have been addressed.

---

> > ### Author Response · Authors · 2024-08-14
> >
> > Thank you for the comments and for raising the score in recognition of our efforts to address them.

---

### Official Review · Reviewer_SKBz · 2024-07-05

**Soundness:** 3
**Presentation:** 3
**Contribution:** 3
**Rating:** 6
**Confidence:** 3

**Summary:**

1. The paper introduces NeuralThink, a novel deep thinking architecture designed to efficiently and consistently extrapolate learned algorithms from smaller problems to larger ones.
2. Unlike previous deep thinking methods, NeuralThink can be applied to both same-size problems (where input and output sizes are the same) and different-size problems (where input and output sizes differ).
3. The architecture consists of three main components:
   - A recurrent module that iteratively processes input information at different scales
   - A processing module that aggregates previously processed information
   - A curriculum-based training scheme to improve extrapolation performance
4. NeuralThink outperforms prior state-of-the-art deep thinking approaches in:
   - Extrapolation capability: Consistently executing learned algorithms on larger problems
   - Training efficiency: Learning algorithms from smaller problems
   - Parameter efficiency: Requiring fewer parameters than other approaches
5. The authors introduce a set of novel different-size tasks to evaluate their method.

**Strengths:**

The paper is easy to follow and experiments nicely presented. The proposed model seems to outperform (one) related work method and results seem quite impressive.

**Weaknesses:**

Since the authors introduce novel benchmarks, the very positive experimental results need more ablations. Please provide hyperparameter scan and their results on baselines.

**Questions:**

See Weaknesses.
Can you provide a more detail comparison between DeepThink and your model. What happens generally for both of these models if you scale them? It would be nice to see when NeuralThink breaks? Afaiu DeepThink was going for the recurrent module to see if larger inputs could be solved by running the network longer in depth. Your model works quite differently afaik and I would again highlight these differences in detail.

**Limitations:**

I would welcome a dedicated limitation section.

---

> ### Author Rebuttal · Authors · 2024-08-06
>
> We express our gratitude to the reviewer for their comments and the effort spent on reviewing our paper. We will now tackle the weaknesses (W), questions (Q), and limitations (L) mentioned:
>
> + **(W1a) Ablations present in the paper:** We provide in the original paper additional ablation studies in Appendix C, due to the lack of space in the main body of the text. In particular we show:
>   + Ablation on the progressive loss parameter of DeepThink: Appendix C.3.
>   + Ablation on the model size of NeuralThink: Appendix C.4.
>   + Ablation on different types of recurrent networks for NeuralThinkL Appendix C.5.
>   + Ablation on layernorm, dropout and projection head: Appendix C.6.
>
>   In the updated version of the paper (beginning of Section 5.3) we have emphasized the existence of these additional ablation studies.
>
> + **(W1b) Hyperparameter scan details on baselines**: In the general rebuttal comment document we show the results of a hyperparameter scan in the GoTo environment for the NeuralThink training hyperparameters. The results show that NeuralThink is robust to hyperparameter changes, with exception of the gradient clip that requires proper tuning. The results further show the significance of using weight decay and both forms of dropout regularization together. In this work we tried to use our model with minimum hyperparameter tuning, keeping the same hyperparameters for the majority of the different-sized environments and using the same hyperparameters for the same-sized tasks proposed by the DeepThink model, as shown in Appendix B.3.
> + **(Q1, Q4) Comparison between DeepThink and our model:** We provide in Appendix B.6. of the original paper an extended discussion on the differences between our model and DeepThink. We have clarified the existence of this comparison in line 129 of the updated version of the paper. Moreover, we also highlight in Appendix C.7. how our model outperforms DeepThink on extrapolation to very-large input sizes (256x256 and 512x512).
> + **(Q2) What happens if you scale the models:** We provide in Appendix C.4. of the original paper an ablation study on the effect of the model size for the performance of NeuralThink. The results show that decreasing the model size results in a decrease in the performance of NeuralThink, yet still outperforms the previous state-of-the-art. In the updated version of the paper (beginning of Section 5.3) we have emphasized the existence of this ablation study.
> + **(Q3) When does NeuralThink break?** In Figures 5 and 6 we highlight that our model struggles to extrapolate when we significantly reduce the size of the input data provided to the model during training. We believe that learning to extrapolate from very small input sizes is a very interesting direction for future work (as mentioned in line 282).
> + **(L1) Dedicated limitations section:** Due to lack of space in the main body, we have created a dedicated Appendix section for limitations, merging it with the previous Appendix B.2. “Changes We Tried That Did Not Improve NeuralThink” and mentioned the existence of this Appendix in line 282 of the updated version of the paper.

---

> > ### Comment · Reviewer_SKBz · 2024-08-11
> > **Thank you!**
> >
> > Thank you for the additional data. I will raise my score to 6. Many thanks!

---

> > > ### Author Response · Authors · 2024-08-14
> > >
> > > Thank you for the discussion and for recognizing our efforts by raising the score.

---

### Official Review · Reviewer_tqxp · 2024-07-08

**Soundness:** 4
**Presentation:** 3
**Contribution:** 3
**Rating:** 7
**Confidence:** 3

**Summary:**

The authors propose a new architecture that improves on the Deep Thinking (DT) architecture by Bansal et al, 2022. It replaces the recurrent ResNet block with a convolutional LSTM+layernorm. They also use a curriculum-based learning schema. This improves the model's extrapolation capability significantly. The authors test their model on identical tasks to those of Bansal et al, and show that they outperform the DT model on all of them, often significantly.

**Strengths:**

The authors propose a strong model that outperforms the baseline in all tasks they test for. Their model is more general than the baseline, allowing generalization also in tasks whose output grid is of a different size compared to the input. The model doesn't suffer from overthinking and does not require special regularization.

**Weaknesses:**

The writing can be improved: the authors do not clearly state what is novel in their architecture in the main text. They should emphasize the exact differences to the literature more clearly: e.g by stating that the difference in their architecture and DT is replacing the ResNet by a convolutional LSTM.

Figures with the double meaning for the color are confusing (Fig 5 and Fig. 6). Also, for Fig 6, there are 3 options but just 2 numbers for each color.

The exact details of the architecture are not entirely clear. It would be better if the authors could describe their architecture with equations, at least in the appendix.

The authors are using LSTMs but do not cite [1].

[1] Hochreiter et al, 1997:  Long Short-Term Memory

**Questions:**

It would be nice to see an ablation on the layernorms in Tab. 4.

In line 168, the authors write "consider the best accuracy obtained by the models at any iteration". Is this also done for NeuralThink? It should not suffer from overthinking, right?

Why is think maze more difficult than the thick one, especially for the baselines?

**Limitations:**

It is not clear how these architectures can transfer from simple grid world problems to some more real-world applications. The authors could discuss this more in the paper.

---

> ### Author Rebuttal · Authors · 2024-08-06
>
> We thank the reviewer for the comments and for the time spent reviewing our paper. We now address the weaknesses (W), questions (Q) and limitations (L) pointed out by the reviewer:
>
> + **(W1) Authors do not clearly state what is novel in their architecture in the main text.** Due to lack of space in the main body, we have an extended discussion on the differences between NeuralThink and DeepThink in Appendix B.6. Nonetheless, we have improved the description of the novelty of our model in line 100 of the updated version of the paper.
> + **(W2) Color labels of  Fig. 5 and Fig. 6 are confusing:** To improve our figures we modified the labels of the colors to “very small”, “small”, “medium” and “large”, and defined these training sizes for each task in Appendix.
> + **(W3) The exact details of the architecture are not entirely clear:** We can describe our architecture as follows: for same size tasks, given an input image $o$, we pass the input through a single layer ConvLSTM ($q$) to obtain the next hidden state ($h\_1$ and cell state ($c\_1$), $h\_1, c\_1 \= q(o,h\_0,c\_0) $. The initial hidden ($h\_0$) and cell ($c\_0$) states are initialized at zeros. The recurrent process is given by feeding back the previous hidden and cell states to the LSTM, $h\_t, c\_t \= q(o,h\_{t-1},c\_{t-1})$. After a fixed size of recurrent iterations, we can obtain the model prediction at timestep $t$ by passing the hidden state $h\_t$ through the processing module. The processing module is composed of three convolutional layers with 3x3 kernels denoted by $W\_1,W\_2,W\_3$ and ReLU activation ($\\gamma$). As such, we perform $p\_t \= W\_3 \\ast \\gamma(W\_2 \\ast \\gamma(W\_1 \\ast h\_t)) $, and a final softmax activation ($\\sigma$), $\\hat{y}\_t \= \\sigma(p\_t)$. For different size tasks, before the softmax we perform a global maxpool operation ($G$) that reduces the processing output height and width to 1, $\\hat{y}\_t \= \\sigma(G(p\_t))$. In the updated version of the paper we added this description to Appendix B.6.
> + **(W4) The authors use LSTMs but do not cite the original article:** We have updated the paper to include the citation of the original LSTM paper.
> + **(Q1) It would be nice to see an ablation on the layernorms in Tab. 4\.** In the original version of the paper we included the ablation on layernorm in Table 15 of Appendix C6, due to lack of space in the main body of the paper. In the updated version of the paper we have emphasized the existence of additional ablation studies available in Appendix, in the beginning of Section 5.3.
> + **(Q2a) Is the best accuracy obtained by the models at any iteration used in NeuralThink?** Yes, we use this evaluation methodology for all methods.
> + **(Q2b) Does NeuralThink suffer from overthinking?** In Appendix B.1. we empirically show that NeuralThink does not suffer from overthinking, across all evaluation tasks. We have updated the text in line 213 to clarify this point.
> + **(Q3) Why is the thin maze more difficult than the thick one, especially for the baselines?** We believe that the difficulty arises from the size of the receptor field in the convolutional layers required to perceive the environment. The original DeepThink architecture uses 5 convolutional layers with a kernel size of 3 for its recurrent module. This means that at each recurrent step, the network is capturing  information from 5 adjacent pixels in each direction (i.e., a square of 11x11 pixels). This fact makes it harder for the model to generalize from smaller problems as the model has to learn an iterative algorithm from a  square of 11 pixels at each iterative step. On the other hand, the ConvLSTM used by NeuralThink only captures information from the next adjacent input pixel (or squares 3x3 pixels). This design choice allows NeuralThink to learn general iterative algorithms from smaller sizes.
> + **(L1) It is not clear how NeuralThink can work on real-world applications:** We have evaluated NeuralThink on both existing benchmarks of DeepThinking models as well as on novel benchmark scenarios that we contribute for different-sized tasks. We have also explored the use of our model in sequential decision-making tasks, such as the DoorKey environment of the MiniGrid suite in Section 5.4, which is still a standard environment for the evaluation of modern reinforcement learning agents (see \[1a-1b\]).
>
> References:
> \[1a\] Nikulin, Alexander, et al. "XLand-minigrid: Scalable meta-reinforcement learning environments in JAX." *arXiv preprint arXiv:2312.12044* (2023).
> \[1b\] Pignatelli, Eduardo, et al. "NAVIX: Scaling MiniGrid Environments with JAX." *arXiv preprint arXiv:2407.19396* (2024).

---

> > ### Comment · Reviewer_tqxp · 2024-08-12
> >
> > I would like to thank the authors for their detailed answers. Given those, I'm raising my score.
> >
> > I would appreciate it if the authors could include the full set of equations describing their model in the appendix of the final version of the paper. Also, point (W1) from the answer to the reviewer dmPv in the main paper would add clarity to the paper.

---

> > > ### Author Response · Authors · 2024-08-14
> > >
> > > Thank you for the suggestions (which we will add to the final version of the paper), the overall discussion and for recognizing our efforts with the score raise.

---

### Author Rebuttal · Authors · 2024-08-06

Dear Reviewers,

We thank all the reviewers for the constructive and interesting questions and suggestions.

We have added a pdf with the additional hyperparameter scan requested by Reviewer **SKBz**, that highlights the robustness of our method to changes in hyperparameters.

Additionally, we replicate here the results of the comparison between our method and LocRNN \[1\], requested by Reviewer **dmPv**, that shows that our method significantly outperforms LocRNN in both extrapolation performance and efficiency.

| Extrapolation accuracy (%) | 1S-Maze | GoTo | Pong | Doorkey |
| :---- | :---- | :---- | :---- | :---- |
| NeuralThink | 100.00 \+- 0.00  | 100.00 \+- 0.00  | 100.00 \+- 0.00  | 100.00 \+- 0.00  |
| LocRNN \[1\] | 87.65 \+- 9.13 | 82.56 \+- 17.73 | 94.08 \+- 8.81 | 86.02 \+- 11.75 |

| Number of parameters (in Millions) | 1S-Maze | GoTo | Pong | Doorkey |
| :---- | :---- | :---- | :---- | :---- |
| NeuralThink | 0.231 | 0.231 | 0.230 | 0.231 |
| LocRNN \[1\] | 0.236 | 0.236 | 0.236 | 0.236 |

| Computational complexity (in gigaMACs) | 1S-Maze | GoTo | Pong | Doorkey |
| :---- | :---- | :---- | :---- | :---- |
| NeuralThink | 4.12 | 9.76 | 9.76 | 9.76 |
| LocRNN \[1\] | 4.33 | 10.25 | 10.25 | 10.25 |


Please let us know if our comments address all the weaknesses and questions pointed out by the reviews or if you require further clarification.

\[1\] Adaptive recurrent vision performs zero-shot computation scaling to unseen difficulty levels. NeurIPS 2023\.

---

### Decision · Program_Chairs · 2024-09-25

**Decision:**

Accept (poster)

**Comment:**

This paper introduces a deep learning approach that generalizes across reasoning-related tasks of different sizes, including asymmetrical tasks. The reviewers agree this is a useful contribution and should be accepted. I accordingly recommendation acceptance. However, as a requirement for acceptance, the term "deep thinking", which represents a gross anthropomorphism of deep learning, **must be removed from this paper in every place it occurs** and both DeepThink and NeuralThink **must be retitled** (the AC will check the camera-ready to make sure that these edits are made). The term "overthinking" is permissible since it is a term for which there is no easy synonym, but other uses of "think" should be eliminated. While the term "deep thinking" was used sparingly in a prior paper, that is not an excuse to perpetuate it beyond acknowledging the prior paper - there are plenty of other terminology options that can be introduced without violating best practices in deep learning communication.